# Computing the relative binding affinity of ligands based on a pairwise binding comparison network

Jie Yu[1,2,3,11], Zhaojun Li[4,5,11], Geng Chen[1,6,7], Xiangtai Kong[1,6], Jie Hu[8], Dingyan Wang[1,3], Duanhua Cao[1,9], Yanbei Li[1,6,7], Ruifeng Huo[8], Gang Wang[1,6], Xiaohong Liu[5], Hualiang Jiang[1,6,8], Xutong Li ⬚[1,6] ✉, Xiaomin Luo ⬚[1,6] ✉ & Mingyue Zheng ⬚[1,6,10] ✉

Structure-based lead optimization is an open challenge in drug discovery, which is still largely driven by hypotheses and depends on the experience of medicinal chemists. Here we propose a pairwise binding comparison network (PBCNet) based on a physics-informed graph attention mechanism, specifically tailored for ranking the relative binding affinity among congeneric ligands. Benchmarking on two held-out sets (provided by Schrödinger and Merck) containing over 460 ligands and 16 targets, PBCNet demonstrated substantial advantages in terms of both prediction accuracy and computational efficiency. Equipped with a fine-tuning operation, the performance of PBCNet reaches that of Schrödinger's FEP+, which is much more computationally intensive and requires substantial expert intervention. A further simulation-based experiment showed that active learning-optimized PBCNet may accelerate lead optimization campaigns by 473%. Finally, for the convenience of users, a web service for PBCNet is established to facilitate complex relative binding affinity prediction through an easy-to-operate graphical interface.

AlphaFold2, which appeared in the 14th round of the Critical Assessment of protein Structure Prediction (CASP), is believed to have solved the half-century-old problem of predicting a protein structure from its primary sequence. This breakthrough has ushered in a new era in structure-based drug design[1]. Recently, the Critical Assessment of Computational Hit-finding Experiments (CACHE), a public benchmarking project, has garnered attention from the computational chemistry community and pharmaceutical industry for enhancing small-molecule hit-finding algorithms[2]. However, the hit-to-lead optimization process is still largely driven by hypotheses and depends on the experience of medicinal chemists. Lead optimization aims to design ligands with higher binding affinity while maintaining other properties[3–5]. During optimization, a congeneric series of ligands is generated that generally share the same core structure and differ only

[1]Drug Discovery and Design Center, State Key Laboratory of Drug Research, Shanghai Institute of Materia Medica, Chinese Academy of Sciences, Shanghai, China. [2]School of Information Science and Technology, Shanghai Tech University, Shanghai, China. [3]Lingang Laboratory, Shanghai, China. [4]College of Computer and Information Engineering, Dezhou University, Dezhou City, China. [5]Development Department, Suzhou Alphama Biotechnology Co., Ltd, Suzhou City, China. [6]University of Chinese Academy of Sciences, Beijing, China. [7]School of Pharmaceutical Science and Technology, Hangzhou Institute for Advanced Study, UCAS, Hangzhou, China. [8]School of Chinese Materia Medica, Nanjing University of Chinese Medicine, Nanjing, Jiangsu, China. [9]Innovation Institute for Artificial Intelligence in Medicine of Zhejiang University, College of Pharmaceutical Sciences, Zhejiang University, Hangzhou, Zhejiang, China. [10]State Key Laboratory of Pharmaceutical Biotechnology, Nanjing University, Nanjing, Jiangsu, China. [11]These authors contributed equally: Jie Yu, Zhaojun Li. ✉e-mail: lixutong@simm.ac.cn; xmluo@simm.ac.cn; myzheng@simm.ac.cn

in some substituent groups. The extensive optimization space for a lead, spanning hundreds to thousands of compounds, necessitates substantial resources for experimental evaluations[6,7]. Consequently, developing in silico predictive tools is important to expedite drug discovery. By minimizing the number of design-make-test-analyze cycles, these tools facilitate the attainment of compounds possessing desired affinity and property profiles.

In recent decades, many relative binding free energy (RBFE) simulation methods have been proposed for lead optimization, benefiting from improved force fields and sampling algorithms. For example, free energy perturbation (FEP) is a widely used alchemical method[8] that is achieving remarkable accuracy on specific systems that is nearing 1 kcal mol$^{-1}$ (ref. 9). However, FEP also suffers from several limitations, such as depending on the process of system preparation for its accuracy[10], being limited by considerable computational cost[9] and being limited to a maximum number of changes between ligands. Another category of RBFE simulation method involves end-points sampling[11], such as the molecular mechanics generalized Born surface area (MM-GB/SA)[12,13]. End-points sampling methods reduce the computational requirements, but their performance is also compromised. In summary, despite the high accuracy of RBFE simulation methods, their complicated preparation process, limited molecule throughput and low allowance for changes between molecules hinder their practical usage in quickly navigating the optimization space of lead molecules.

In recent years, some artificial intelligence (AI) models designed for guiding lead optimization have emerged[14–16]. Inspired by RBFE simulation methods, Jiménez-Luna et al. proposed a convolutional Siamese neural network (SNN), called DeltaDelta[15], to directly determine the RBFE between two bound ligands. One advantage of SNN is that it directly determines the RBFE, which eliminates the systematic error derived from the absolute binding free energies (ABFEs). Another advantage is its ability to factor in information from both input ligands, incorporating their structural differences and commonalities. However, DeltaDelta has yet to take full advantage of the SNN architecture. Specifically, DeltaDelta first predicts the ABFE of two inputted compounds, and then directly uses the difference of the predicted ABFE as the final RBFE prediction for loss calculation. This approach does not consider the association between the two inputs (pairwise separability)[17]. DeltaDelta showed relatively poor outcomes in retrospective lead optimization campaigns without fine-tuning. McNutt et al. recently proposed a multitask convolutional SNN model[16]. Their approach involves using the explicit differences between the representations of two inputted ligands as the molecular-pair representation. The potential assumption is that features that are common to two ligands are irrelevant to predicting their difference, which is obviously unreasonable in RBFE predictions. Moreover, they used the prediction of the ABFE as one of the auxiliary tasks, potentially reintroducing the noise originally eliminated by RBFE prediction. Consequently, compared with DeltaDelta, their models did not show substantial performance gains.

In summary, developing an efficient and accurate method to guide lead optimization is an urgent need. To this end, we propose a pairwise binding comparison network (PBCNet) based on a physics-informed graph attention mechanism that is specifically tailored for ranking the relative binding affinity among a congeneric series of ligands. Several physical-oriented modeling strategies are introduced, considering that the formation of intermolecular interactions always follows strict geometric rules[18]. Based on our interpretation studies, we found that a relatively high attention score assigned to protein–ligand atom pairs may indicate a more significant interaction. Additionally, PBCNet focuses on molecular substructures that can form intermolecular interactions.

PBCNet has been evaluated in terms of the error and correlation between the predicted and experimental binding affinities. Benchmarking results show that our model substantially outperformed all baselines except FEP+. Furthermore, with a small amount of fine-tuning[19] data, PBCNet is comparable to Schrödinger's FEP+, but with substantially less computational cost. An ideal model should also have the ability to enrich key high-activity compounds from a batch of structural analogs. We built a benchmark to test whether our model can identify 'leading' compounds, and the results indicate that, on average, PBCNet can accelerate lead optimization projects by 473%. Finally, PBCNet has been deployed in the cloud, and the corresponding web service is accessible at https://pbcnet.alphama.com.cn/index.

## Results

### Model structure

The framework of PBCNet is shown in Fig. 1. It consists of three parts: (1) the message-passing phase, (2) the readout phase and (3) the prediction phase. The input of PBCNet is a pair of pocket–ligand complexes in which the ligands are structural analogs and the parts comprising the pockets are entirely identical. The amino-acid residues of the protein for which the minimum distance for the ligand is less than or equal to 8.0 Å are kept as the protein pocket. The message-passing phase is designed to obtain node-level representations. First, the graph convolutional network (GCN)[20] is applied to update the atom representations of the protein pocket alone. Then, the updated protein pocket is combined with the two ligands by building edges between pairs of atoms less than 5.0 Å apart. A well-designed message-passing network (detailed in the Methods) is then used to transmit information across the molecule graphs. Finally, we remove the pocket from the molecular graphs and only retain the ligands. The goal of the readout phase is to obtain the molecular representations (graph-level). In this phase, molecular representations of the ligands ($\mathbf{x}^{(i)}$ and $\mathbf{x}^{(j)}$ in Fig. 1) are computed by an Attentive FP[21] readout operation. Then, the molecular-pair representations ($\tilde{\mathbf{x}}^{(i,j)}$ in Fig. 1) are obtained by equation (7) in the Methods. In the prediction phase, molecular-pair representations are learned by optimizing the losses of two tasks: (1) the predictions of affinity differences and (2) the probabilities that the affinity of ligand $i$ is greater than that of ligand $j$ by two independent branches of three-layer feedforward neural networks (see section Model training and fine-tuning process).

In the inference process, we only need to provide docking poses of a pair of structurally similar small molecules to the same protein to obtain the predicted relative binding affinity. A more detailed description of the model framework, and the difference between the Siamese network and traditional networks are also demonstrated in the Methods.

### Performance of PBCNet

**Zero-shot learning.** First, we analyzed the zero-shot performance of PBCNet on the two held-out test sets (FEP1 and FEP2 sets, see section Benchmark dataset for performance assessment), and selected Schrödinger's FEP+ (ref. 9), Schrödinger's Glide SP[22], MM-GB/SA[11], as well as four AI-based models (DeltaDelta[15], Default2018 (ref. 16), Dense[16] and PIGNet[23]) as baselines. The general idea of zero-shot learning is to transfer the knowledge contained in the training instances to the task of testing instance prediction[24]. This evaluation is designed to simulate the early stage of a lead-optimization campaign, where there is always a lack of compounds with known activity. For each test series we randomly selected one ligand as the reference ligand to infer the absolute binding affinities of the remaining ligands (see section Mathematical formulation), and this process was repeated ten times to avoid randomness. The performances of all methods on the FEP1 and FEP2 sets are summarized in Supplementary Data 1 and 2, respectively. Pearson's correlation coefficient ($R$), Spearman's rank correlation coefficient ($\rho$) and the pairwise root-mean-square error (r.m.s.e.$_{pw}$) are used here (see section Determination of model performance). For PIGNet, the results were calculated using its officially reported code and weights. For other baselines, we utilized performance metrics as detailed in their respective original literature.

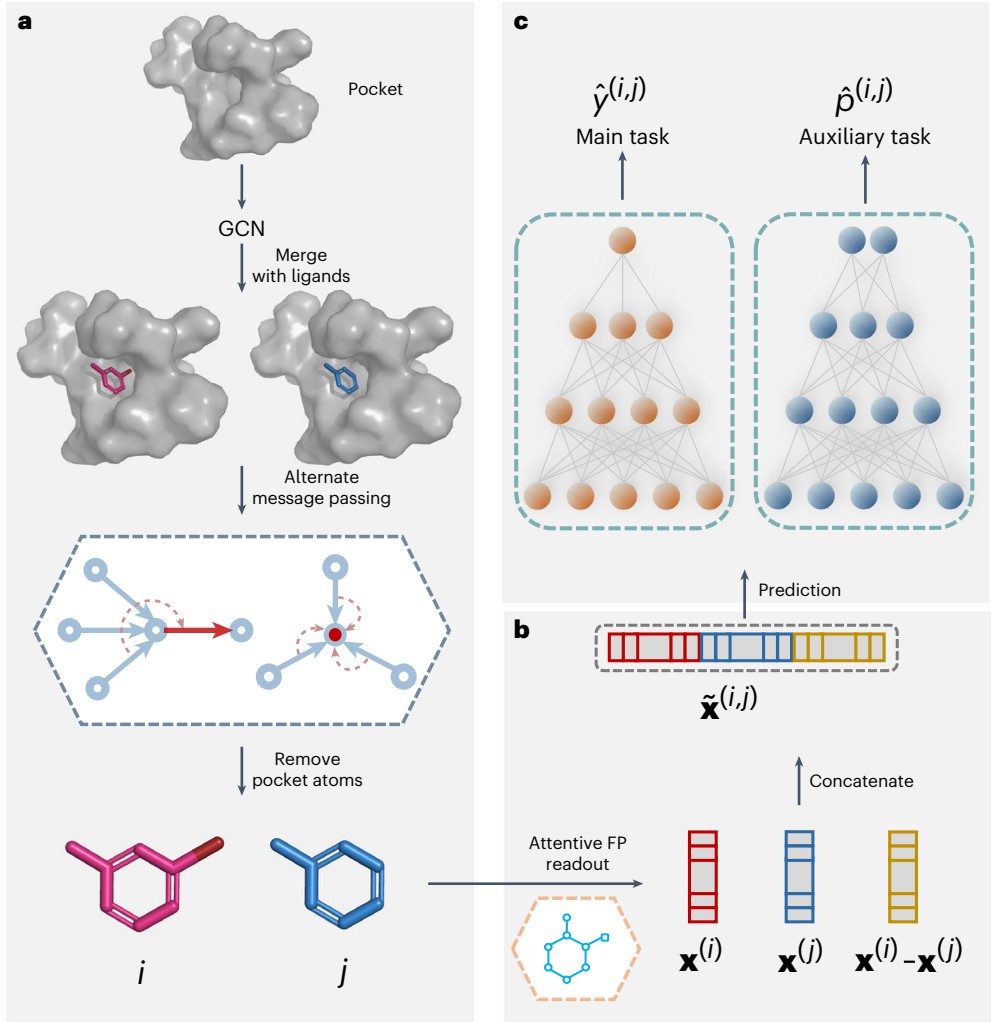

**Fig. 1 | The framework of PBCNet. a**, Message-passing phase. This phase is used to realize the mutual information interaction between the ligands (in red and blue) and the protein pocket (in gray), and obtain node-level representations of the ligands. **b**, The readout phase obtains the molecular representations (graph-level) and realizes the information interaction of the pair of ligands. The red and blue nodes represent the graph-level representations of ligand $i$ and ligand $j$, respectively ($\mathbf{x}^{(i)}$ and $\mathbf{x}^{(j)}$), and the yellow nodes present the difference of the two graph-level representations, $\mathbf{x}^{(i)} - \mathbf{x}^{(j)}$. The molecular-pair representations $\tilde{\mathbf{x}}^{(i,j)}$ are obtained by splicing between the three. **c**, In the prediction phase, molecular-pair representations are learned by optimizing the losses of two tasks: (1) predictions of affinity differences $\hat{y}^{(i,j)}$ and (2) the probabilities ($\hat{p}^{(i,j)}$) that the affinity of ligand $i$ is greater than that of ligand $j$ by two independent branches of three-layer feedforward neural networks.

The results show that the performance of PBCNet is substantially better than that of all baselines except FEP+, meaning that PBCNet is the best of all high-throughput methods mentioned here. Moreover, the accuracy of PBCNet on the FEP1 set has achieved 1.11 kcal mol$^{-1}$, which is very close to 1 kcal mol$^{-1}$, and it also achieves the lowest average r.m.s.e.$_{pw}$ (1.49 kcal mol$^{-1}$) on the FEP2 set. Supplementary Fig. 1 visualizes the model predictions, demonstrating a strong alignment between the predicted ΔpIC$_{50}$ values (ΔpIC$_{50}$ is the difference between the pIC$_{50}$ values of two ligands, pIC$_{50}$ is the negative logarithm of IC$_{50}$ in molar concentration and IC$_{50}$ means 50% inhibitory concentration, which is a type of binding affinity. Please see section Training dataset and data balance) and the corresponding experimental values across the majority of the test series.

We also find that PBCNet is robust, with more stable performance across all testing series compared with other high-throughput baseline methods. This is evident from the Spearman's rank correlation coefficient; PBCNet shows correlations of over 0.30 in all test series, whereas other high-throughput baseline methods show a more fluctuating $\rho$, such as Glide SP (CKD2, $\rho = -0.36$; Tyk2, $\rho = 0.79$). This phenomenon reflects the good generalization ability of PBCNet.

Then, we can also observe that the performance of PBCNet on the FEP1 set is better than that on the FEP2 set, possibly due to the several out-of-domain samples in the FEP2 set. As a model for lead optimization, PBCNet is designed to infer the activity differences of structural analogs, which always generate high molecule similarities. To be closely consistent with the application scenario, the training set is composed of molecule pairs whose Tanimoto similarity scores are higher than 0.6 (ref. 25). Figure 2a shows the relationship between the model accuracy and molecule similarity, and an obvious negative correlation can be observed. It is not a surprise to notice the similarity-dependent performance of PBCNet, because identifying molecules with different structures is more relevant to virtual screening than lead optimization. Correspondingly, the methods and models designed for virtual screening are always poor at lead optimization, such as Glide and PIGNet, which have been evaluated here. We further counted the proportions of ligand pairs with different similarity scores in the FEP1 and FEP2 sets (Fig. 2b). Figure 2b shows that the proportion of molecule pairs with a Tanimoto similarity score of less than 0.6 in the FEP2 set are substantially higher than that in the FEP1 set (70.4% versus 54.4%), which may lead to the performance differences of our model on the

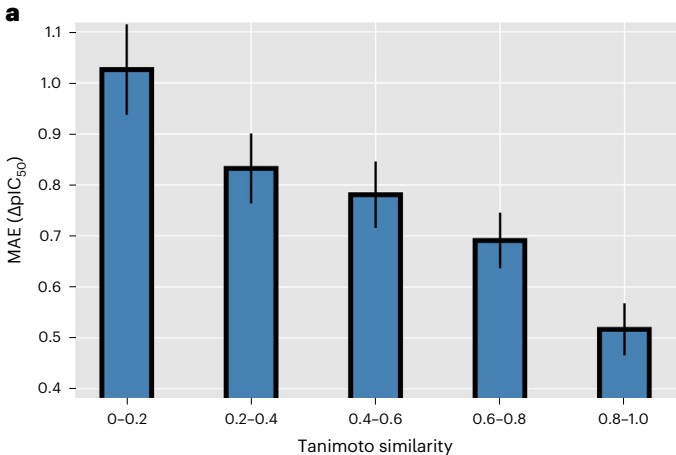

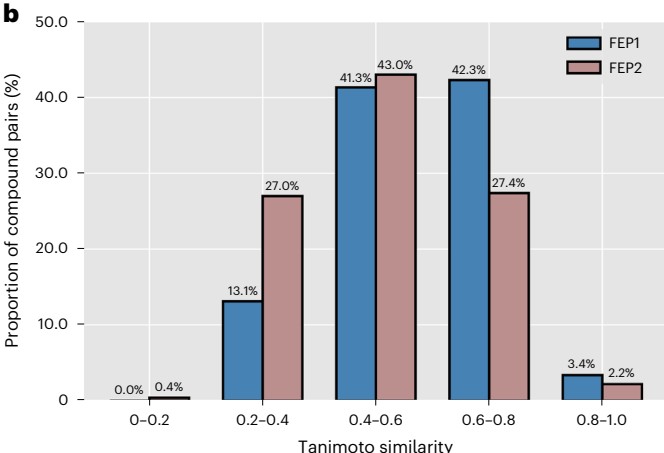

**Fig. 2 | Performance analysis of PBCNet on the FEP1 and FEP2 sets. a**, Bar plot showing the change in model accuracy with pairwise molecule similarity. We split all pairwise samples in both test sets, ordered by Tanimoto similarity scores in five bins ($x$ axis), and calculated the mean absolute errors (MAEs) for each bin ($y$ axis). The error bars represent 0.1 times the standard deviation (bin 0–0.2, $n = 18$; bin 0.2–0.4, $n = 1,567$; bin 0.4–0.6, $n = 3,071$; bin 0.6–0.8, $n = 2,404$;

bin 0.8–1.0, $n = 195$). **b**, Bar plot showing the proportions of ligand pairs ($y$ axis) with different Tanimoto similarity scores ($x$ axis) in the FEP1 and FEP2 sets. The proportion of molecules pairs with a Tanimoto similarity score less than 0.6 in the FEP2 set are substantially higher than in the FEP1 set (70.4% versus 54.4%), and all pairs with a Tanimoto similarity score of less than 0.2 are from the FEP2 set.

FEP1 and FEP2 sets. However, PBCNet's ranking performance on the FEP2 set still surpassed all the baselines, except for FEP+. Given this, we may conclude that PBCNet should be of practical value for guiding lead-optimization projects.

Finally, we also find our model is highly robust to small changes in ligand poses (specific information is provided in Supplementary Section 1).

**Few-shot learning.** The reason why we assumed the ranking ability of PBCNet to be inferior to that of FEP+ is because of the ability of FEP+ to sample various binding conformations. Other methods, except MM-GB/SA, only use a single snapshot, which leads to less comprehensive information about the molecular binding process. However, PBCNet has two advantages over FEP+ in a real-world application. First, PBCNet is not limited by molecule throughput, allowing for comprehensive exploration of lead optimization. According to public information[9], running FEP+ for four perturbations per day requires eight commodity Nvidia GTX-780 graphics processing units (GPUs). In contrast, PBCNet takes only 0.9 s to calculate one perturbation by use of a commodity Nvidia V100 GPU. Through a rough performance conversion, PBCNet is ~100,000 times faster than FEP+. The second advantage is PBCNet's flexibility. During a lead-optimization campaign, the binding affinity data newly generated can be used to fine-tune PBCNet. Few-shot learning[19] is used to achieve this. For each test congeneric series, we randomly selected several ligands (~2–10) as fine-tuning ligands with known binding affinity, which also serve as reference ligands in the inference phase. The remaining ligands are still the ligands to be tested (referred to as the new testing series). We repeat the above process ten times to avoid randomness.

The performances of the fine-tuned models on the new testing series are summarized in Supplementary Data 3 and Fig. 3. Figure 3 shows that the few-shot learning strategy substantially improves the performance of PBCNet, and the performance increases with the number of fine-tuning ligands. Supplementary Table 1 shows that the performances of the fine-tuned PBCNet on the new and original testing series are similar. This suggests that the performance improvement is not due to the bias resulting from the reduced length of the test series. This consistency is also essential for comparing the fine-tuned PBCNet and FEP+ under existing conditions. We find that, after fine-tuning, PBCNet's ranking ability is comparable to that of FEP+.

For example, PBCNet fine-tuned with four ligands even outperformed FEP+ in terms of Spearman's rank correlation coefficient on the FEP1 set (0.724 versus 0.720).

## Using PBCNet to accelerate lead optimization

In this section we test whether our model can efficiently identify high-activity compounds in a close-to-real-world lead-optimization scenario by comparing the order of model selection to the experimental order of synthesis, similar to the study of Jiménez–Luna and others[15]. We use active learning (AL)[26], an uncertainty-guided algorithm, to intelligently prioritize sample acquisition. Data acquisition was simulated as iterative selection from each chemical series, with PBCNet as the active learner. In each series, the compound displaying the highest activity was used as the target ligand that needs to be identified. In cases where multiple compounds hold the same highest activity, we prioritized the earliest synthesized among them as the target ligand. In the first iteration, the earliest synthesized compound in each chemical series was chosen as the reference ligand, and activity values were evaluated across the remaining compounds. Subsequently, three ligands with the highest predictive values were selected. If the target ligand was not among these three, they become new reference ligands for the next iteration. In the second iteration, four existing reference ligands were paired to form a fine-tune set for refining PBCNet. Both the predicted activity values and uncertainties (equations (10) and (11) in the Methods) of the remaining ligands were evaluated by the fine-tuned PBCNet. This evaluation guided the prioritization of three ligands, according to the predefined sampling method. This iteration was repeated until the target ligand was successfully identified.

We adopted three sampling methods with different settings (see section The sample method for simulation-based experiment). Results for this simulation-based benchmark are presented in Supplementary Data 4. We find that the strategies taking uncertainty into consideration are superior to the purely exploitation-oriented one, and the model-oriented as well as user-oriented strategies do not exhibit an obvious performance difference. The model-oriented AL strategy is selected as the representative for further comparison, and three metrics are used and computed as follows:

$$\text{Advantage order} = \text{Experimental order} - \text{Model selection order} \tag{1}$$

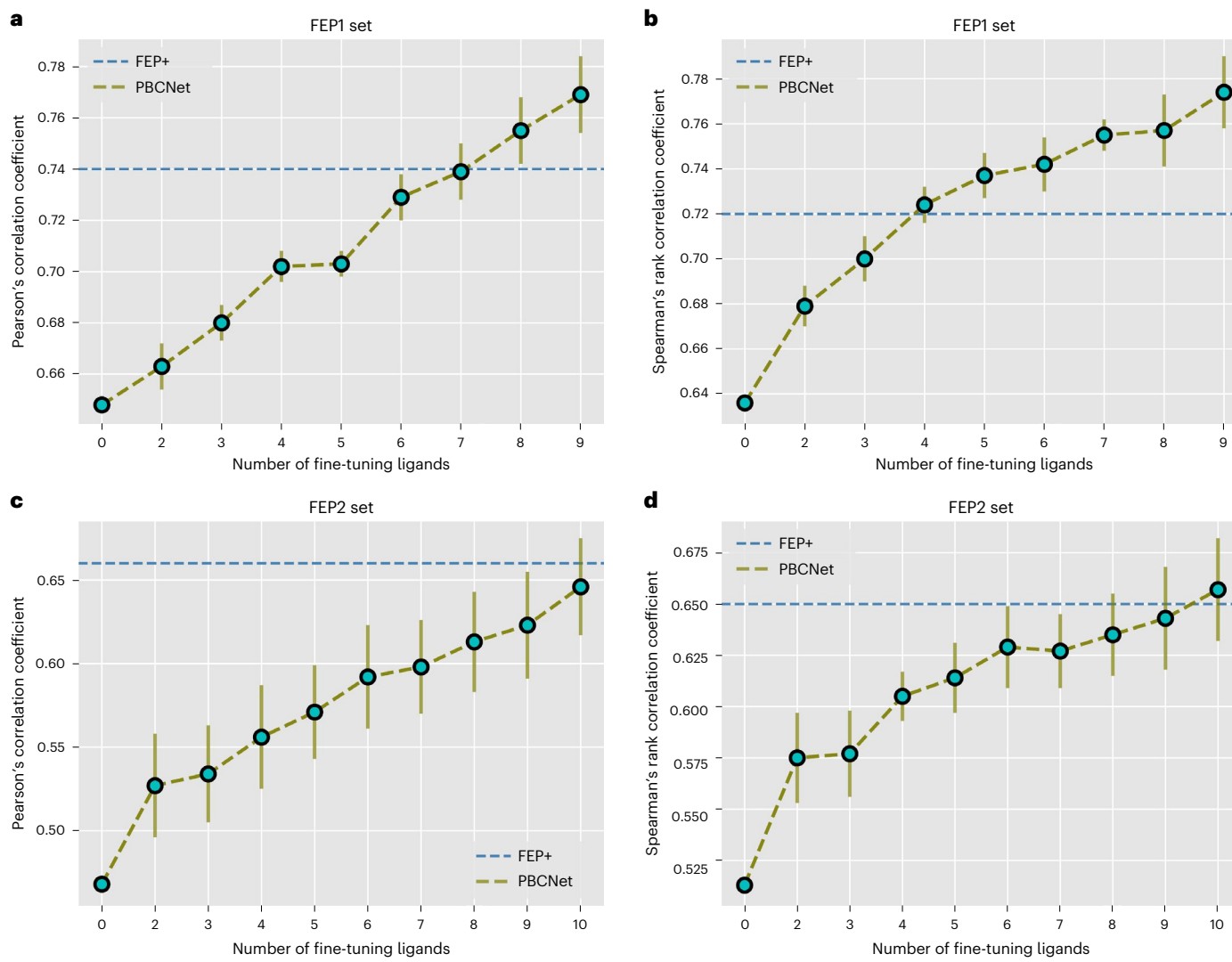

**Fig. 3 | Change in performance of PBCNet as the number of fine-tuning ligands varies.** The *x* axis of each subplot indicates the number of fine-tuning ligands, and the *y* axis indicates the model ranking performance. Blue dashed lines indicate the performance of FEP+. Error bars represent the standard deviation of the ranking performance for ten independent runs ($n = 10$). From the graphs we can see that the performance of PBCNet increases as the number of fine-tuning compounds increases.

$$\text{Advantage ratio}$$
$$= \frac{\text{Experimental order} - \text{Model selection order}}{\text{Number of ligands}} \times 100\% \qquad (2)$$

$$\text{Efficiency improvement ratio}$$
$$= \frac{\text{Experimental order} - \text{Model selection order}}{\text{Model selection order}} \times 100\% \qquad (3)$$

The 'advantage ratio' represents the theoretical percentage of resources saved when utilizing PBCNet for guiding lead optimization, compared to not using it. The 'efficiency improvement ratio' represents the increase in efficiency when completing a compound optimization project before and after using PBCNet, assuming that a project ends after obtaining the most active compound.

In six out of nine datasets, AL-equipped PBCNet is able to attain the compound with the highest affinity faster than its experimental order. On average, it accelerated the lead-optimization projects by ~473%, while also achieving an ~30% reduction in resource investment. Surprisingly, for the BCL6, sEH and AAK1 targets, the compounds with

the highest affinity were found by PBCNet in the first iteration without the fine-tuning operation. We compared our results to the baseline MM-GB/SA, which was implemented using the Schrödinger Prime MM-GBSA with default settings. The results, presented in Supplementary Table 2, demonstrate that PBCNet consistently outperforms MM-GB/SA across all evaluated metrics. Overall, the results are very promising and suggest that PBCNet could be successfully applied in a prospective scenario to accelerate lead optimization.

### Model interpretability analysis

**Atom level.** Given PBCNet's impressive performance, it is valuable to investigate how the model makes predictions. Because PBCNet is attention-based, the attention score between a pair of atoms can be seen as a measure of importance. A strong model should assign high scores to atom pairs forming key intermolecular interactions. To illustrate this, we performed a case study on two different ligands in the FEP1 set, focusing on identifying hydrogen bonds[27], which are crucial and common intermolecular interactions.

We first computed the intermolecular interactions between the ligands and proteins with Schrödinger2020-4. Because the positions of the hydrogen atoms depended heavily on the program used to add

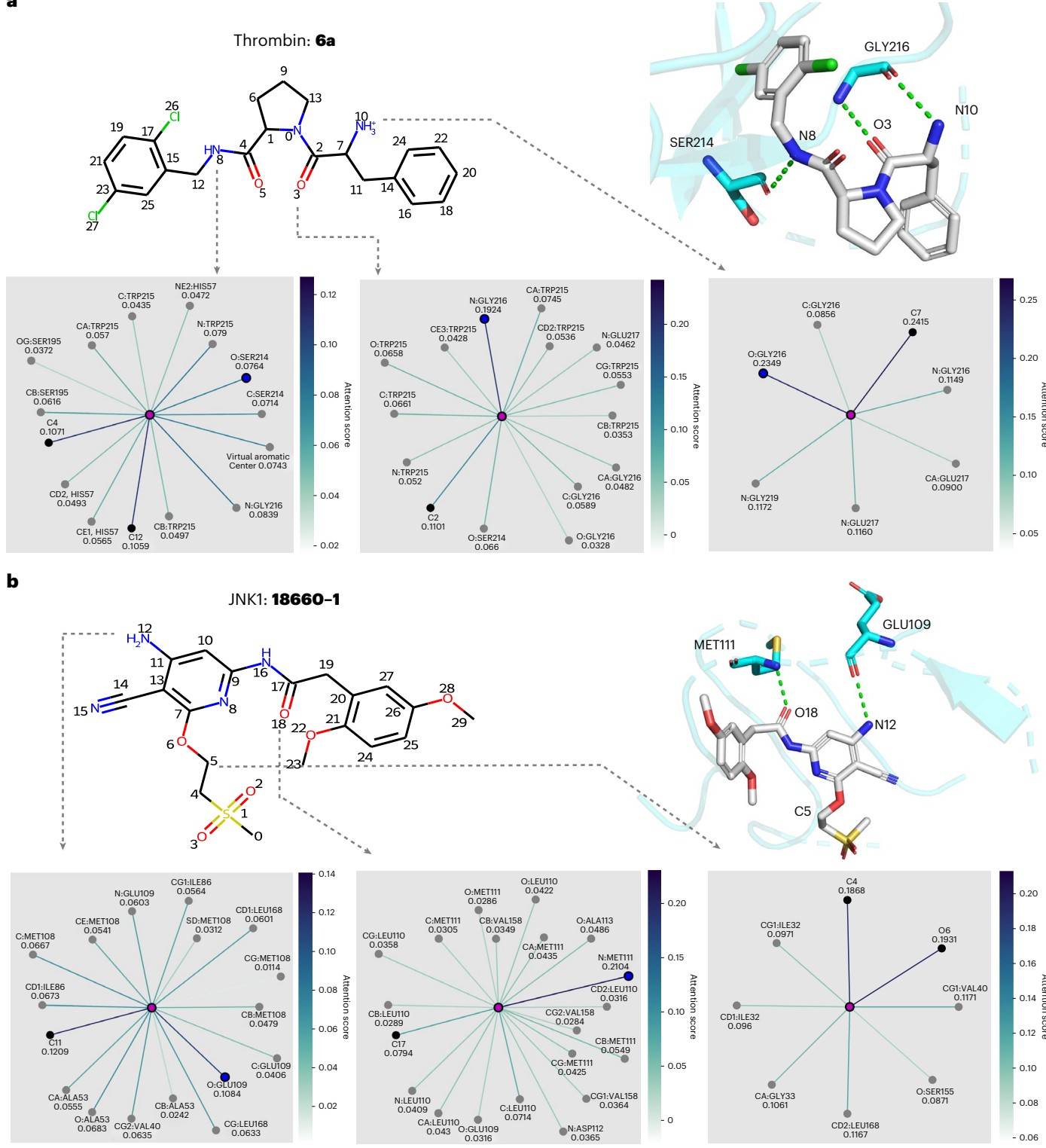

**Fig. 4 | Node-level interpretability analysis results of PBCNet on two ligands. a,b**, A thrombin inhibitor **6a** (**a**) and a JNK1 inhibitor **18660-1** (**b**). The molecular structure, three-dimensional hydrogen-bond visualization graphs and attention visualization graphs are shown for comparison. In each attention visualization graph, the ligand atom (referred to as target atom) is denoted by a purple dot, indicated by an arrow and is involved in the formation of hydrogen bonds. Other dots denote the neighbor atoms of the target atom. The black dots represent the ligand atoms (including the virtual aromatic nodes in the ligand structure) covalently linked with the target atom, the gray ones represent the protein pocket atoms (including the virtual aromatic nodes in the protein structure) linked with the target atom by virtual distance edges and the dot in blue denotes the protein pocket atom that forms the hydrogen bond with the target atom. The color of the edges is coded based on their attention score, and an edge with a dark color is favorable for protein–ligand binding.

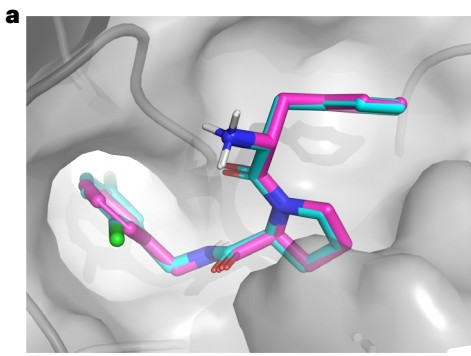

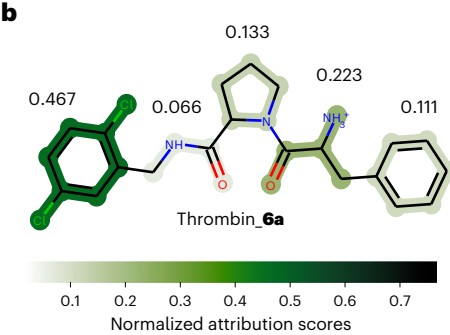

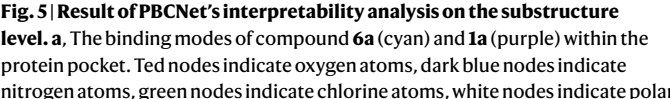

**Fig. 5 | Result of PBCNet's interpretability analysis on the substructure level. a**, The binding modes of compound **6a** (cyan) and **1a** (purple) within the protein pocket. Ted nodes indicate oxygen atoms, dark blue nodes indicate nitrogen atoms, green nodes indicate chlorine atoms, white nodes indicate polar hydrogen atoms and the rest of the nodes indicate carbon atoms. **b**, Visualization of the analysis: each substructure is color-coded according to its normalized attribution scores.

hydrogens, we did not take them into account. For hydrogen-bond donors, we selected the heavy atoms covalently linked with hydrogen atoms for further analysis. We then extracted the attention weights, generated in the last layer of the Distance-aware edge to node block (Methods), of the atoms involved in the formation of hydrogen bonds. The results of these operations are illustrated in Fig. 4, and the inter-molecular interactions computed by Schrödinger are summarized in Supplementary Table 3.

Compound **6a** from the thrombin series forms three hydrogen bonds with the target at the 3, 8 and 10 positions (Fig. 4a). We found that the hydrogen bonds formed at the 3 and 10 positions are high-lighted. The covalent bonds are also emphasized. This is consistent with a chemical prior that the chemical environment of a ligand atom is largely determined by its covalently linked atoms and the protein atoms involved in key intermolecular interactions. It reveals that PBCNet is able to capture key intermolecular interactions. The com-puted hydrogen bond at the 8 position is not emphasized, unlike its counterparts at the 3 and 10 positions, possibly due to the relatively weaker hydrogen-bond donor nature of the amide-donor hydro-gen atom[28]. Compound **18660-1** from the JNK1 series forms two hydrogen bonds with the target at the 12 and 18 positions (Fig. 4b). As expected, all of them are highlighted. Moreover, the carbon atom of **18660-1** at the 5 position, which does not form any key intermo-lecular interaction (computed by Schrödinger), was selected as a negative sample. We can clearly see that only covalent bonds are assigned relatively high attention scores, while the attention scores of the virtual distance bonds are small and uniform in value. The above results all reflect the rationality of the prediction basis of our model.

**Substructure level.** Medicinal chemists prefer to investigate molecular properties in terms of chemically meaningful fragments rather than individual atoms[29]. Therefore, we extended our analysis to include substructure-level interpretability.

In this analysis, we employed the substructure mask explanation (SME) methodology, as recently proposed by Wu and others[29]. We assume that the model's prediction value for a compound is denoted as $\hat{y}$. Then, the compounds are split into substructures using the BRICS method. Sequentially, the hidden representations of the atoms of each substructure are masked during the readout phase, yielding the cor-responding prediction value $\hat{y}_{\mathrm{sub}_i}$ where the subscript $\mathrm{sub}_i$ represents the $i$th substructure. When the predicted value represents the com-pound's activity, we consider that a greater decrease in $\hat{y}_{\mathrm{sub}_i}$ compared to $\hat{y}$ indicates that the corresponding substructure plays a more crucial role in the model's prediction. Thus, the attribution scores used to

quantify the importance of each substructure are defined by the fol-lowing equation:

$$\mathrm{Attribution}_{\mathrm{sub}_i} = \hat{y} - \hat{y}_{\mathrm{sub}_i} \qquad (4)$$

and we normalize the attribution scores to normalized attribution scores (Attribution_N) within a range of 0 and 1, according to

$$\mathrm{Attribution\_N}_{\mathrm{sub}_i} = \frac{\mathrm{Attribution}_{\mathrm{sub}_i}}{\sum_{i=1}^{N} \mathrm{Attribution}_{\mathrm{sub}_i}} \qquad (5)$$

where $N$ is the number of substructures.

Here, we take compound **6a** from the thrombin system as a case study, using compound **1a** as a reference ligand to illustrate PBCNet's activity prediction for compound **6a** (Fig. 5a). Compound **6a** was seg-mented into seven substructures using the BRICS method, with the amide group being divided into two distinct substructures. To provide a more intuitive representation for medicinal chemists, we manually merged the amide group as a whole (Supplementary Table 4). The visualization is presented in Fig. 5b.

As shown, we found that Sub$_4$ and Sub$_1$ (Supplementary Table 4) have the greatest impact on the predictive results. PBCNet is designed to predict the relative binding affinities, which are predominantly derived from the different substructures of a pair of ligands. Sub4, being the part of compound **6a** that structurally deviates from com-pound **1a**, has been emphasized, suggesting that PBCNet indeed cap-tures the structural differences between input ligands. Moreover, as depicted in Fig. 4a, Sub$_1$ forms two hydrogen bonds with the protein, so the emphasizing of Sub$_1$ also implies that PBCNet focuses on key molecular motifs that form intermolecular interactions.

**Ablation experiments**

To enhance the performance of PBCNet, we implemented various strategies, which can be broadly divided into two categories: frame-work-related and knowledge-related. The former includes the SNN architecture and the classification assistance task, while the latter incorporates physical and prior knowledge. To verify whether these strategies really contribute to the model performance improvement, we performed the following ablation experiments on PBCNet.

PBCNet stands out due to its SNN network framework with paired inputs. We constructed a single-input model termed 'Singular PBCNet' to remove the SNN framework. Meanwhile, to verify the effect of pair-wise separability on the SNN framework, we built a pairwise separated model referred to 'Separated PBCNet'. Their frameworks are shown in

Supplementary Fig. 2. We also removed the classification auxiliary task and obtained 'MSE PBCNet'. Note that Singular PBCNet and Separated PBCNet lack the assistance task as they do not use molecular pairs information, and their performance should be compared with MSE PBCNet subsequently. The performance of the ablated models is shown in Supplementary Table 5.

Compared with PBCNet, MSE PBCNet showed a small decrease in performance on both the FEP1 and FEP2 sets (FEP1, 0.636 versus 0.629; FEP2, 0.513 versus 0.488). This aligns with expectations, as the auxiliary task addresses samples with small errors but wrong rankings, which constitute a small fraction of the dataset. Compared with MSE PBCNet, the performance of Singular PBCNet showed a substantial decrease both on the FEP1 set and on the FEP2 set (FEP1, 0.629 versus 0.559; FEP2, 0.488 versus 0.372 (statistically significant)). This result illustrates the advantage of the SNN framework in relative binding affinity prediction. Compared with MSE PBCNet, the performance of Separated PBCNet significantly decreases on the FEP2 set (0.488 versus 0.425). For such results we believe that the ability to consider the structural information of both inputted molecules and their connections simultaneously is crucial for the model performance.

We next removed the distance information, angle information and aromatic information, separately. The performance of the ablated PBCNet is shown in Supplementary Table 5. After removing any of the knowledge-related strategies, the performance of PBCNet decreases on both the FEP1 and FEP2 sets, especially the distance information. This phenomenon indicates that all three knowledge-related strategies contribute to the performance of PBCNet.

## Discussion

AI has gained prominence in solving scientific problems by incorporating domain-specific knowledge into its modeling. PBCNet is an example of this integration of physical knowledge into its framework. However, there are still avenues for improvement. First, although PBCNet shows substantial predictive advancements over prior attempts, its zero-shot performance is lower than that of Schrödinger's FEP+. Therefore, capturing protein conformational changes prompted by ligand binding, just like FEP+, remains an ongoing pursuit to improve model accuracy. Second, the underlying assumption of this study is that similar ligands exhibit similar binding modes. Therefore, extreme cases, where highly similar ligands bind to the protein with entirely different binding modes, may pose challenges for PBCNet's handling capabilities. Furthermore, PBCNet still relies on medicinal chemists for molecule design and molecular docking binding poses generation. A direct-shot pipeline that integrates molecular generation, docking and optimization, could circumvent cumulative errors in the process of lead optimization.

In the future, we will continue to refine our modeling strategies to enhance PBCNet's predictive performance by considering the alterations of protein conformation and ligand pose. Simultaneously, we will also try to combine PBCNet with deep molecular generative models to streamline the automated design of high-potency molecules.

## Methods
### Mathematical formulation

In traditional modeling protocols (single-input modeling methods), suppose we are given a training set with $N$ samples (protein–ligand complexes from the same congeneric series) $\mathcal{D}=\{\mathbf{x}^{(i)}, y^{(i)}\}_{i=1}^{N}$. Here, $\mathbf{x}^{(i)} \in \mathbb{R}^m$ represents the feature vector of an input, $m$ means its dimension and $y^{(i)} \in \mathbb{R}$ is a real-valued property (pIC$_{50}$ here). $\mathcal{M}$ is a deep learning-based regression model parameterized by weights $\boldsymbol{\theta}$ and trained on $\mathcal{D}$, and $y^{(i)} = \mathcal{M}(\mathbf{x}^{(i)}; \boldsymbol{\theta})$ represents the prediction result of $\mathcal{M}$ for $\mathbf{x}^{(i)}$.

For Siamese models, however, these concepts are subject to slight change. First, $N$ training samples are paired with each other to form $\binom{N}{2}$ paired training samples, and tuple $p$ is used to index them:

$$p \in \{(i,j) \mid 1 \le i < j \le N\} \tag{6}$$

where $i$ and $j$ correspond to indexes of the first and second complex of a paired sample. Then, the feature vector $\tilde{\mathbf{x}}^{(i,j)}$ of a paired sample is dependent on $\mathbf{x}^{(i)}$ and $\mathbf{x}^{(j)}$. Here, $\tilde{\mathbf{x}}^{(i,j)} \in \mathbb{R}^{3*m}$ is constructed by the following equation:

$$\tilde{\mathbf{x}}^{(i,j)} = \mathbf{x}^{(i)} \oplus \mathbf{x}^{(j)} \oplus (\mathbf{x}^{(i)} - \mathbf{x}^{(j)}) \tag{7}$$

where $\oplus$ is the concatenation operation. The label of a paired sample $\tilde{y}^{(i,j)}$ ($\Delta$pIC$_{50}$ here) is calculated according to

$$\tilde{y}^{(i,j)} = y^{(i)} - y^{(j)} \tag{8}$$

Finally, the pairwise training dataset $\mathcal{D}_p = \{\tilde{\mathbf{x}}^{(i,j)}, \tilde{y}^{(i,j)}\}_{1 \le i < j \le N}$ is obtained. $\mathcal{M}_p$ is a Siamese regression model parameterized by weights $\boldsymbol{\theta}_p$ and trained on $\mathcal{D}_p$. $\tilde{y}^{(i,j)} = \mathcal{M}_p(\tilde{\mathbf{x}}^{(i,j)}; \boldsymbol{\theta}_p)$ represents the prediction result of $\mathcal{M}_p$ for $\tilde{\mathbf{x}}^{(i,j)}$.

For an unseen complex $u$ whose feature vector is represented by $\mathbf{x}^{(u)}$, we pair it with every complex in $\mathcal{D}$, which can be seen as a set of reference samples with known binding affinities in the inference phase, to obtain the pairwise test dataset $\{\tilde{\mathbf{x}}^{(i,u)}, \tilde{y}^{(i,u)}\}_{i=1}^{N}$. $\mathcal{M}_p$ is able to output the corresponding $N$ predictions $\{\tilde{y}^{(i,u)}\}_{i=1}^{N}$, and the predicted absolute affinity of $u$ $\{\hat{y}_i^{(u)}\}_{i=1}^{N}$ based on different reference samples can be obtained by the equations

$$
\begin{aligned}
\hat{y}_1^{(u)} &= y^{(1)} - \tilde{y}^{(1,u)} \\
\hat{y}_2^{(u)} &= y^{(2)} - \tilde{y}^{(2,u)} \\
&\vdots \\
\hat{y}_N^{(u)} &= y^{(N)} - \tilde{y}^{(N,u)}
\end{aligned}
\tag{9}
$$

The mean value and variance of $\{\hat{y}_i^{(u)}\}_{i=1}^{N}$ can be deemed the final prediction $\hat{y}^{(u)}$ and uncertainty estimation $\sigma^{2(u)}$ of $u$, respectively (equations (10) and (11)):

$$\hat{y}^{(u)} = \frac{1}{N} \sum_{i=1}^{N} \hat{y}_i^{(u)} \tag{10}$$

$$\sigma^{2(u)} = \frac{1}{N} \sum_{i=1}^{N} \left( \hat{y}^{(u)} - \hat{y}_i^{(u)} \right)^2 \tag{11}$$

### The structure of alternately updated message-passing neural network

A well-designed message-passing neural network (alternately updated message-passing neural network, AU-MPNN) is applied in the message-passing phase (Fig. 1a). Before the detailed introduction of AU-MPNN, some definitions need to be clarified. First, the complex of a ligand and the corresponding protein binding pocket is deemed a directed molecular graph $G$, in which all heavy atoms are treated as nodes ($Nd$), and all covalent bonds are treated as edges ($E$). Moreover, virtual distance edges are built between atom pairs of the ligand and the binding pocket, whose distances are less than or equal to 5.0 Å. Additionally, virtual aromatic nodes are set up for the centroid of each aromatic ring, and virtual aromatic edges are also established between virtual aromatic nodes and the nodes in corresponding aromatic rings. During message passing, all nodes (heavy atom nodes and virtual aromatic nodes) and all edges (covalent bond edges, virtual distance edges and virtual aromatic edges) are equivalent. Finally, the final whole graph $G = \langle Nd, E \rangle$ is constructed. Here, all edges are directed, and an edge $e_{\overrightarrow{uv}}$ indicates that its direction goes from node $a_u$ to node $a_v$. If there is an edge $e_{\overrightarrow{uv}}$ in $G$, $a_u$ is a neighbor node of $a_v$. In the following, $a_v$ is assumed to be the target node whose representation needs to be updated. The set

$V_{nei} = \{a_{u_1}, a_{u_2}, a_{u_3}, \cdots\}$ represents all neighbor nodes of $a_v$, and $a_u$ refers to any neighbor node of $a_v$ (Supplementary Fig. 3a). Correspondingly, the set $UV = \{e_{\vec{u_1 v}}, e_{\vec{u_2 v}}, e_{\vec{u_3 v}}, \cdots\}$ is all incoming edges of $a_v$ (edges that point to $a_v$). Moreover, $e_{\vec{uv}}$ is assumed to be the target edge that needs to be updated. The set $U_{nei} = \{a_{k_1}, a_{k_2}, a_{k_3}, \cdots\}$ represents all neighbor nodes of $a_u$ except $a_v$. The set $KU = \{e_{\vec{k_1 u}}, e_{\vec{k_2 u}}, e_{\vec{k_3 u}}, \cdots\}$ stands for all neighbor edges of $e_{\vec{uv}}$, and $e_{\vec{ku}}$ refers to any neighbor edge of $e_{\vec{uv}}$ (Supplementary Fig. 3a).

The specific architecture of AU-MPNN is shown in Supplementary Fig. 3c. In general, AU-MNPP consists of two phases: (1) distance and angle-aware bond-to-bond blocks and (2) distance-aware bond-to-atom blocks. In the following sections, we will give a detailed introduction for these two phases and the corresponding preparations.

**Initial featurization.** Node and edge features need to be defined before message passing. Here we use a total of 15 types of atomic feature (Supplementary Table 6) and five types of bond feature (Supplementary Table 7) to characterize them and their local chemical environment. Except for atomic mass, explicit valence, implicit valence and van der Waals (vdw) radius, the rest of these features are encoded in a one-hot fashion. Of note is that the feature vectors of virtual nodes and edges are set as zero vectors.

**Initial hidden representations.** Initial node and edge features should be further encoded as their initial hidden representations before the first step of message passing. Taking $a_v$ and $e_{\vec{uv}}$ as examples, we initialize their hidden representations with

$$\mathbf{h}_v^0 = \text{ReLU}\left(W_{i-node} \times \mathbf{x}_v + b_{i-node}\right) \qquad (12)$$

$$\mathbf{x}_{\vec{uv}}' = \text{ReLU}\left(W_{i-edge} \times \mathbf{x}_{\vec{uv}} + b_{i-edge}\right) \qquad (13)$$

$$\mathbf{h}_{\vec{uv}}^0 = \text{ReLU}\left(W_i \times \text{cat}\left(\mathbf{h}_u^0, \mathbf{x}_{\vec{uv}}'\right) + b_i\right) \qquad (14)$$

where $\mathbf{x}_v \in \mathbb{R}^{l_{node}}$ and $\mathbf{x}_{\vec{uv}} \in \mathbb{R}^{l_{edge}}$ are initial features of $a_v$ and $e_{\vec{uv}}$; $\mathbf{h}_v^0 \in \mathbb{R}^m$, $\mathbf{h}_u^0 \in \mathbb{R}^m$ and $\mathbf{h}_{\vec{uv}}^0 \in \mathbb{R}^m$ are initial hidden representations of $a_v$, $a_u$ and $e_{\vec{uv}}$, respectively; $\mathbf{x}_{\vec{uv}}' \in \mathbb{R}^{\frac{m}{2}}$ is an intermediate vector to obtain $\mathbf{h}_{\vec{uv}}^0$; $\text{cat}(\cdot)$ is the concatenate operation; $W_{i-node}$, $W_{i-edge}$ and $W_i$ are learned matrices; and i means 'initial'. This process is visualized in Supplementary Fig. 3b.

**Distance and angle-aware edge-to-edge blocks (DAEE blocks).** The aim of this block is to use the information of the neighbor edges in $KU$ to update the hidden representation of $e_{\vec{uv}}$. For $e_{\vec{uv}}$, the neighbor edges are not equally important. For example, a neighbor edge that stands for a key intermolecular interaction between ligand and protein should be highlighted. Hence, the attention mechanism in GAT[30] is applied here. Moreover, considering that intermolecular interactions are determined by the atomic types and distances, atom pairwise statistical potentials[31] are introduced as an additional attention bias term. Here, the Bayesian field theory-based potentials[32] proposed by Zheng et al. are adopted. Additionally, the degree of the angle between two edges also limits the formation of intermolecular interactions (for example, hydrogen bonds and halogen bonds). Thus, angle information is taken into consideration in computing the attention scores.

The computing process of this block is summarized in Supplementary Fig. 3c (left). First, on each step $l$, the queries of $e_{\vec{uv}}$ ($\mathbf{q}_{\vec{uv}}^l$) and the keys of its any neighbor edge $e_{\vec{ku}}$ ($\mathbf{k}_{\vec{ku}}^l$) are obtained according to

$$\mathbf{q}_{\vec{uv}}^l = W_{q-edge}^l \times \mathbf{h}_{\vec{uv}}^{l-1} + b_{q-edge}^l \qquad (15)$$

$$\mathbf{k}_{\vec{ku}}^l = W_{k-edge}^l \times \mathbf{h}_{\vec{ku}}^{l-1} + b_{k-edge}^l \qquad (16)$$

where $W_{q-edge}^l$ and $W_{k-edge}^l$ are two learned matrices. According to the spatial coordinates of nodes $a_k$, $a_u$ and $a_v$, the degree of angle $\theta_{kuv}$ between $e_{\vec{ku}}$ and $e_{\vec{uv}}$ can be computed. Then, we divide the angles into six angle domains with a cutoff of $\frac{\pi}{6}$ (Supplementary Fig. 3d), and encode them as the corresponding angle embedding. Here, the angle information is fused by extending the original attention mechanism in the GAT with angle-aware attention:

$$\varepsilon_{\vec{uv},\vec{ku}}^l = \mathbf{w}_{edge}^l \cdot \text{LeakyReLU}\left[\mathbf{q}_{\vec{uv}}^l + \mathbf{k}_{\vec{ku}}^l + W_{angle}^l \times \text{Divider}\left(\theta_{kuv}\right)\right] \qquad (17)$$

where Divider is used to map $\theta_{kuv}$ to the located angle domain one-hot vector, $W_{angle}^l$ is a learned matrix, $\mathbf{w}_{edge}^l$ is a learned vector and $\varepsilon_{\vec{uv},\vec{ku}}^l$ is the correlation coefficient of $e_{\vec{ku}}$ and $e_{\vec{uv}}$. After that, atom pairwise statistical potentials are converted as an additional bias term ($p_{k,u}$) to combine distance information:

$$p_{k,u} = \begin{cases} 1 & \text{if } e_{\vec{ku}} \text{ is a covalent bond} \\ 2 \times \log\left(P\left(\text{type}_k, \text{type}_u, \text{dist}_{\vec{ku}}\right)\right) & \text{if } e_{\vec{ku}} \text{ is a virtual bond} \\ 0.8 & \text{if type}_k \text{ or type}_u \text{ is not covered} \end{cases} \qquad (18)$$

$$\varepsilon_{\vec{uv},\vec{ku}}'^l = \varepsilon_{\vec{uv},\vec{ku}}^l + p_{k,u} \qquad (19)$$

$$\alpha_{\vec{uv},\vec{ku}}^l = \frac{\exp\left(\varepsilon_{\vec{uv},\vec{ku}}'^l\right)}{\sum_{e_{\vec{ku}} \in KU} \exp\left(\varepsilon_{\vec{uv},\vec{ku}}'^l\right)} \qquad (20)$$

where $\text{type}_k$ and $\text{type}_u$ are atomic types of $a_k$ and $a_u$; $\text{dist}_{\vec{ku}}$ represents the distance between $a_k$ and $a_u$ (meaning the length of $e_{\vec{ku}}$); $P(\cdot)$ is the mapping function of atom pairwise statistical potentials; $\varepsilon_{\vec{uv},\vec{ku}}'^l$ is the updated correlation coefficient of $e_{\vec{ku}}$ and $e_{\vec{uv}}$; and the final calculated attention score $\alpha_{\vec{uv},\vec{ku}}^l$ reflects how important $e_{\vec{ku}}$ is for $e_{\vec{uv}}$. Then, the message embedding ($\mathbf{m}_{\vec{uv}}^l$) used to update the hidden representation of $e_{\vec{uv}}$ is computed according to:

$$\mathbf{m}_{\vec{uv}}^l = \sum_{e_{\vec{ku}} \in KU} \alpha_{\vec{uv},\vec{ku}}^l \times \mathbf{k}_{\vec{ku}}^l \qquad (21)$$

Finally, the updated hidden representation of $e_{\vec{uv}}$ ($\mathbf{h}_{\vec{uv}}^l$) is acquired by residual connections by the following equation:

$$\mathbf{h}_{\vec{uv}}^l = \text{Res}\left(\text{Res}\left(\mathbf{h}_{\vec{uv}}^{l-1} + W_{edge-2}^l \times \text{ReLU}\left(W_{edge-1}^l \times \mathbf{m}_{\vec{uv}}^l\right)\right)\right) \qquad (22)$$

where $W_{edge-1}^l$ and $W_{edge-2}^l$ are trained parameter matrices, and $\text{Res}(\cdot)$ is the residual connection module (Supplementary Fig. 3e).

**Distance-aware edge-to-node blocks (DEN blocks).** The goal of this block is to use the information of the neighbor nodes in $V_{nei}$ and the incoming edges in $UV$ to update the hidden representation of $a_v$. The computing process of this block is summarized in Supplementary Fig. 3c (right). Similar to DAEE blocks, we also introduce the attention mechanism and additional distance-based bias term. Similarly, the message-passing phase of the DEN block operates according to

$$\mathbf{q}_v^l = W_{q-node}^l \times \mathbf{h}_v^{l-1} + b_{q-node}^l \qquad (23)$$

$$\mathbf{k}_u^l = W_{k-node}^l \times \mathbf{h}_u^{l-1} + b_{k-node}^l \qquad (24)$$

followed by

$$\varepsilon_{u,v}^l = \mathbf{w}_{\text{node}}^l \cdot \text{LeakyReLU}\left(\mathbf{q}_v^l + \mathbf{k}_u^l\right) \quad (25)$$

$$\varepsilon_{uv}'^l = \varepsilon_{uv}^l + p_{u,v} \quad (26)$$

$$\alpha_{u,v}^l = \frac{\exp\left(\varepsilon_{u,v}'^l\right)}{\sum_{a_u \in V_{\text{nei}}} \exp\left(\varepsilon_{u,v}'^l\right)} \quad (27)$$

followed by

$$\mathbf{m}_v^l = \sum_{e_{uv}^\rightarrow \in UV} \alpha_{u,v}^l \times \mathbf{h}_{uv}^l \quad (28)$$

$$\mathbf{h}_v^l = \text{Res}\left(\text{Res}\left(\mathbf{h}_v^{l-1} + W_{\text{node}-2}^l \times \text{ReLU}\left(W_{\text{node}-1}^l \times \mathbf{m}_v^l\right)\right)\right) \quad (29)$$

Note that all the variables here correspond to those in the DAEE blocks.

## Data collection and processing

**Training dataset and data balance.** In this study, the BindingDB protein–ligand validation sets (2020 version)[33] were selected as the original training data source. A total of 1,265 congeneric series were included in the dataset, and, for each series, SMILES (Simplified Molecular Input Line Entry System) of the ligands, PDB IDs of the available cocrystal structures and corresponding binding affinity values were provided by the dataset.

The goal of data processing is to generate docking poses of all the ligands and their corresponding proteins by Glide as the input of our model. SMILES that failed during preparation with RDKit[34] were removed. Binding affinity measurements without values as well as uncertain, for example, qualified data with either the '<' or '>' sign, were discarded. The initial three-dimensional structures of the ligands were constructed using RDKit. Then, the ligands were further preprocessed for docking using the Schrödinger LigPrep module with default parameters. From the protein side, the PDB files were prepared using the Protein Preparation Wizard of the Schrödinger suite, following the default protocol. Resolved water molecules that made more than three hydrogen bonds to ligand or receptor atoms were kept, and the structure was centered using the co-crystallized ligand as the center of the receptor grid generated for each protein structure. According to the statistics, 843 (out of 1,265) series possessed multiple available PDB files. For each of these congeneric series, a cross-docking experiment (taking the observed binding site from one protein–ligand complex and docking a different ligand into the site) was carried out to obtain the protein structure with the best pose prediction accuracy for further investigation[35]. After the pretreatment, the docking was performed using the Glide module in Schrödinger with default parameters, and at most 100 poses per ligand can be written out. Medicinal chemists have long recognized that ligands from the same chemical series tend to bind a given protein in similar poses[36]; therefore, a key step of pose selection was performed here. For each series, the maximum common substructure (MCS) of each ligand and the co-crystallized ligand was extracted first. Then, the r.m.s.d. of each pose of a ligand and the experimentally determined pose of the co-crystallized ligand in the MCS moiety were calculated, and if the r.m.s.d. was within 2.0 Å, the corresponding pose (referred to as the acceptable pose) will be considered to share the same binding mode with the co-crystallized ligand. When there are multiple acceptable poses of a ligand, the pose with the highest glide score is selected as the final pose. When we cannot obtain the acceptable pose of a ligand through docking, however, the ligand will be discarded to ensure data quality. The above operations associated with Schrödinger were implemented with the 2020-4 version and by the Schrödinger

Python API. The Numpy[37], Pandas[38] and scikit-learn[39] packages were used for data processing. Matplotlib[40] was used for visualization.

A total of 1,007 (out of 1,265) series with $IC_{50}$ affinity values were extracted (this was the unit with most data available), containing a diverse set of targets. The $IC_{50}$ affinity values were then log-converted to avoid target scaling issues ($pIC_{50} = -\log_{10} IC_{50}$). Accordingly, the $pIC_{50}$ difference ($\Delta pIC_{50}$) between a pair of ligands from the same congeneric series was chosen as the model prediction target here. Twenty-six congeneric series including only one ligand (could not form ligand pairs) and ten congeneric series containing the same protein and ligand as the hold-out test congeneric series (detailed in the next section) were also removed. As a result, there is no overlap in the test congeneric series with the training datasets. Finally, we obtained 971 congeneric series with an average of ~34 ligands per series.

Additionally, we found that the labels of the training data were normally distributed, and most of them were concentrated in the area of [−1, 1] (Supplementary Fig. 4a), which would easily lead to overfitting (a model is able to achieve a low training error as long as the model predicts the mean value of the training labels). Thus, we balanced the training data by undersampling the samples in the high-density regions and oversampling the samples in the low-density regions to alleviate this problem. The label distribution of the balanced training dataset is shown in Supplementary Fig. 4b. The final training dataset consists of 0.6 million pairwise samples.

**Benchmark dataset for performance assessment.** Datasets provided by Wang et al.[9] and Schindler et al.[6] were chosen as the held-out test sets and used to benchmark the performance of different methods for lead optimization in this study. Wang et al. provide eight congeneric series (referred to as the FEP1 set) on different targets with experimentally validated binding free energy $\Delta G$ values and corresponding evaluation statistics of FEP calculations. We converted $\Delta G$ values to the $pIC_{50}$ range assuming non-competitive binding, generating the following equation for conversion:

$$pIC_{50} \approx -\log_{10}\left(e^{\frac{\Delta G}{RT}}\right) \quad (30)$$

where $R = 1.987 \times 10^{-3}$ kcal $K^{-1}$ $mol^{-1}$ is the gas constant, $T = 297$ K is the thermodynamic temperature and e = 2.718 is the Euler number. Schindler et al. also provided eight congeneric series (referred to as the FEP2 set) with pharmaceutically relevant targets, all with experimentally measured binding affinities ($IC_{50}$ values). Compared with the FEP1 set, the congeneric series in the FEP2 set contains changes in net charge and the charge distribution of molecules as well as ring openings and core hopping. For each series, we also log-converted the labels and paired the ligands as we did for the training data.

**Benchmark dataset for simulation-based experiment.** Apart from the assessment of model accuracy and model ranking ability on the whole congeneric series, we still intend to test whether our model is able to efficiently identify key high-activity compounds in a close-to-real-world lead-optimization scenario, by retrospectively comparing the order of model selection to the experimental order of synthesis, similar to Jiménez-Luna and others[15]. On this basis, we constructed a benchmark consisting of nine recently published datasets[41–49] with available cocrystal structures and pharmaceutically relevant targets. All series were processed as we did for the training data. The information (for example, protein name and PDB ID) about the benchmark is summarized in Supplementary Table 8.

## Determination of model performance

We include three different metrics used to determine the performance of the predictive models. Pearson's correlation coefficient ($R$) and Spearman's rank correlation coefficient ($\rho$) are used to evaluate the

ranking ability, and r.m.s.e._pw is used to assess the accuracy of the predictive models.

Note that PBCNet requires at least one reference complex to infer the predictive affinities of other test samples and calculate the corresponding $R$ and $\rho$. As a result, the test process was repeated ten times independently and the reference complex of each test process was randomly selected to simulate the uncertainty in real applications.

R.m.s.e. is defined as

$$\text{R.m.s.e.} = \sqrt{\frac{1}{N}\sum_{u=1}^{N}\left(y^{(u)} - \hat{y}^{(u)}\right)^2} \tag{31}$$

where $u$ corresponds to a test sample (a protein–ligand complex here); $y^{(u)}$ and $\hat{y}^{(u)}$ are the true label and prediction results of the test sample, respectively; and $N$ is the total number of test samples. R.m.s.e._pw is defined as

$$\text{R.m.s.e.}_{\text{pw}} = \sqrt{\frac{1}{N}\sum_{u=1}^{N}\left(\tilde{y}^{(i,u)} - \hat{y}^{(i,u)}\right)^2} \tag{32}$$

where $(i, u)$ corresponds to a paired test sample composed of a test complex and any reference complex (from the same congeneric series), and $\tilde{y}^{(i,u)}$ and $\hat{y}^{(i,u)}$ are the true label and prediction results of the paired test sample, respectively. Note that here we use r.m.s.e._pw to evaluate the accuracy of the models. The reason for this is that we use experimental affinities of reference complexes to achieve the conversion of $\hat{y}^{(u)}$ and $\hat{y}^{(i,u)}$ (equation (8)), as Wang et al. and Schindler et al. did in their studies. Additionally, r.m.s.e._pw in the kcal mol$^{-1}$ and pIC$_{50}$ units of our model are reported to compare with baseline models from different studies.

### Model training and fine-tuning process

As discussed in the Model structure section, a hybrid loss function is deployed in the training process with equation (33):

$$\text{Loss}_{\text{total}} = \text{Loss}_{\text{MSE}} + \alpha\text{Loss}_{\text{entropy}} \tag{33}$$

where $\alpha$ is a factor controlling the balance between the two types of loss, which can be seen as a hyperparameter. Here, $\alpha$ is set as 1, Loss$_{\text{MSE}}$ is the loss of mean-square-error loss function, Loss$_{\text{entropy}}$ is entropy loss and Loss$_{\text{total}}$ is final loss. The aim of the introduction of entropy loss is to penalize the predictions with low errors but completely wrong ranking. For example, it is difficult for the regression loss function to penalize a sample with a label of 0.1 and a predicted value of −0.1 due to its low MSE value, but this can be effectively realized by the classification loss function. Additionally, the ranking information contained in the hidden representation of a paired sample may be further reinforced by the auxiliary task to improve the ranking ability of PBCNet.

Hyperparameter optimization was performed by grid research on the training data with inter-congeneric series fivefold cross-validation. Considering the considerable number of training samples, 0.25 epochs was set as the unit of early stopping. In the final training process, the model is trained using a batch size of 96 samples for 5.75 epochs with a learning rate of 5e$^{-7}$.

In the fine-tuning phase, we did not perform the auxiliary task of PBCNet. PBCNet was fine-tuned using a batch size of 30 samples for 10 epochs with a learning rate of 1e$^{-5}$.

### Sample method for simulation-based experiment

The sampling method we define here is as follows:

$$a = \begin{cases} \hat{y} & N_{\text{ite}} = 1 \\ \hat{y} + \beta\sigma^2 & N_{\text{ite}} \geq 2 \end{cases} \tag{34}$$

where $\hat{y}$ and $\sigma^2$ are the predicted activity value and uncertainty, $a$ is the acquisition score, $N_{\text{ite}}$ is the number of iterations and $\beta$ is a user-defined parameter adjusting the exploration–exploitation trade-off. Different values of $\beta$ correspond to three different situations:

- $\beta$ is equal to zero. It is a purely exploitation-oriented AL scenario where the users do not take uncertainty into consideration.
- $\beta$ is more than zero (a hybrid AL scenario). This sampling strategy is model-oriented or in favor of 'exploration'. Samples with greater uncertainty have a higher possibility to be selected (meaning more structure–activity relationship will be explored), so that the fine-tuned model's applicability domain may be expanded and the model is expected to give more reliable predictions in the followed iterations.
- $\beta$ is less than zero. This sampling strategy is user-oriented or in favor of 'exploitation'. In a real-world scenario, the compounds with the highest predicted activity values will be selected for further experimental verification. However, compounds with greater uncertainty are more likely to be overestimated. Given this point, users may tend to treat uncertainty as a penalty term to ensure the data quality in this iteration.

The strategies mentioned above are all simulated in our work ($\beta = 0, 2, -2$, respectively), and six independent runs with different random seeds are conducted.

### Statistics and reproducibility

The $P$ values to test for differences in ablation experiments were calculated using a two-sided Wilcoxon signed rank test. The sample size for each analysis was determined by the maximum number of eligible samples available in the respective datasets. The study design did not require blinding. The model's performance testing involves randomness in the selection of test and reference samples. To mitigate its impact, we conducted multiple repeated experiments using controlled random seed settings ($n = 10$). To reproduce the primary results of this research, refer to the analytical pipeline available at https://doi.org/10.5281/zenodo.8275244 (ref. 50).

### Reporting summary

Further information on research design is available in the Nature Portfolio Reporting Summary linked to this Article.

## Data availability

The unprocessed training data are from BindingDB source and can be found at https://www.bindingdb.org/validation_sets/index.jsp. The test datasets used in this study are available at https://doi.org/10.5281/zenodo.8275244 (ref. 50). Source data are provided with this paper.

## Code availability

The source code for PBCNet is available in the Code Ocean software capsule: https://doi.org/10.24433/CO.1095515.v2 (ref. 51).

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

## Acknowledgements

This work was supported by the National Natural Science Foundation of China (T2225002, 82273855 to M.Z.; 82130108 to X. Luo; 82204278 to X. Li), Lingang Laboratory (LG202102-01-02 to M.Z.), the National Key Research and Development Program of China (2022YFC3400504 to M.Z.), China Postdoctoral Science Foundation (2022M720153 to X. Li), SIMM-SHUTCM Traditional Chinese Medicine Innovation Joint Research Program (E2G805H to M.Z.), Shanghai Municipal Science and Technology Major Project, and the open fund of state key laboratory of Pharmaceutical Biotechnology, Nanjing University, China (KF-202301 to M.Z.).

## Author contributions

J.Y., M.Z., X. Luo, X. Li, H.J. and D.W. designed the research study. J.Y. developed the method and wrote the code. G.C., X.K., J.H., D.C., G.W., R.H. and Y.L. performed the analysis. J.Y., M.Z. and X. Luo wrote the paper. Z.L., J.Y. and X. Liu developed the web service. All authors read and approved the manuscript.

## Competing interests

The authors declare no competing interests.

## Additional information

**Correspondence and requests for materials** should be addressed to Xutong Li, Xiaomin Luo or Mingyue Zheng.

# Reporting Summary

## Statistics

For all statistical analyses, confirm that the following items are present in the figure legend, table legend, main text, or Methods section.

| n/a | Confirmed | |
|---|---|---|
| ☐ | ☒ | The exact sample size (*n*) for each experimental group/condition, given as a discrete number and unit of measurement |
| ☐ | ☒ | A statement on whether measurements were taken from distinct samples or whether the same sample was measured repeatedly |
| ☐ | ☒ | The statistical test(s) used AND whether they are one- or two-sided<br>*Only common tests should be described solely by name; describe more complex techniques in the Methods section.* |
| ☐ | ☒ | A description of all covariates tested |
| ☒ | ☐ | A description of any assumptions or corrections, such as tests of normality and adjustment for multiple comparisons |
| ☐ | ☒ | A full description of the statistical parameters including central tendency (e.g. means) or other basic estimates (e.g. regression coefficient) AND variation (e.g. standard deviation) or associated estimates of uncertainty (e.g. confidence intervals) |
| ☐ | ☒ | For null hypothesis testing, the test statistic (e.g. *F*, *t*, *r*) with confidence intervals, effect sizes, degrees of freedom and *P* value noted<br>*Give P values as exact values whenever suitable.* |
| ☒ | ☐ | For Bayesian analysis, information on the choice of priors and Markov chain Monte Carlo settings |
| ☒ | ☐ | For hierarchical and complex designs, identification of the appropriate level for tests and full reporting of outcomes |
| ☐ | ☒ | Estimates of effect sizes (e.g. Cohen's *d*, Pearson's *r*), indicating how they were calculated |

*Our web collection on statistics for biologists contains articles on many of the points above.*

## Software and code

Policy information about availability of computer code

| Data collection | Python 3.7, Rdkit 2021.03.02, Schrödinger2020-4, pandas v1.1.5, numpy 1.23.5, scikit-learn 1.0.2 |
|---|---|
| Data analysis | python 3.7, Rdkit 2021.03.02, pandas v1.1.5, numpy 1.23.5, scikit-learn 1.0.2, matplotlib 3.3.4 |

For manuscripts utilizing custom algorithms or software that are central to the research but not yet described in published literature, software must be made available to editors and reviewers. We strongly encourage code deposition in a community repository (e.g. GitHub). See the Nature Portfolio guidelines for submitting code & software for further information.

## Data

Policy information about availability of data

All manuscripts must include a data availability statement. This statement should provide the following information, where applicable:

- Accession codes, unique identifiers, or web links for publicly available datasets
- A description of any restrictions on data availability
- For clinical datasets or third party data, please ensure that the statement adheres to our policy

The unprocessed training data is from BindingDB source and can be found at https://www.bindingdb.org/validation_sets/index.jsp. The test datasets used in this study are available at https://doi.org/10.5281/zenodo.8275244, where all molecule and protein files of FEP1 and FEP2 sets could be found. For benchmark dataset of the simulation-based experiment, all molecules and protein files also can be found at https://doi.org/10.5281/zenodo.8275244, including the following PDB files: 7OZY, 7Q7R, 3TGM, 7SUF, 7P4K, 7NWK, 7U9Y, 7RJ7, and 5V3Y.  Source data for Fig. 2-4 and Fig. 5b is available with this manuscript.

# Human research participants

Policy information about <u>studies involving human research participants and Sex and Gender in Research.</u>

| | |
|---|---|
| Reporting on sex and gender | n/a |
| Population characteristics | n/a |
| Recruitment | n/a |
| Ethics oversight | n/a |

Note that full information on the approval of the study protocol must also be provided in the manuscript.

# Field-specific reporting

Please select the one below that is the best fit for your research. If you are not sure, read the appropriate sections before making your selection.

☒ Life sciences ☐ Behavioural & social sciences ☐ Ecological, evolutionary & environmental sciences

For a reference copy of the document with all sections, see nature.com/documents/nr-reporting-summary-flat.pdf

# Life sciences study design

All studies must disclose on these points even when the disclosure is negative.

| | |
|---|---|
| Sample size | 1) In this study, the BindingDB protein-ligand validation sets (2020 version) were selected as the original training data source. A total of 1265 congeneric series were included in the dataset, and, for each series, SMILES (Simplified Molecular Input Line Entry System) of the ligands, PDB IDs of the available cocrystal structures, and corresponding binding affinity values were provided by the dataset. Finally, we got 971 congeneric series with an average of about 34 ligands per series.<br>2) For the performance analysis of PBCNet on FEP1 and FEP2 sets (Fig. 2A), the sample size for each analysis was determined by the maximum number of eligible samples available in the respective datasets (bin 0.0-0.2: n=18, bin 0.2-0.4: n=1567, bin 0.4-0.6: n=3071, bin 0.6-0.8: n=2404, bin 0.8-1.0: n=195).<br>3) For the results in 'The performance of PBCNet on $\Delta pIC_{50}$ calculation' section, we all performed 10 independent runs with different random seed (n=10). Since our model requires a reference molecule, the choice of the reference molecule affects the performance of the model. In order to more fully validate the model performance, we conducted 10 independent experiments. The reason for setting n=10 is that there is one test series which contains only 11 molecules and n=10 is sufficient to make a full assessment of the model performance.<br>4) For the results in 'Using active learning in PBCNet to accelerate lead optimization ' section, we all performed 6 independent runs with different random seed(n=6). The process involves fine-tuning the model. Randomness in the AI model training process affects the training of the model, which is unavoidable. In this experiment we set up the random seed in order to control the randomness and ensure that the relevant experimental results can be reproduced.<br>5) In the model robustness validation experiments, the sample size is determined by the maximum number of ligand poses that can be produced by the docking software in a regular process (Bace: n=7, CDK2: n=3, JNK1: n=3, MCL1: n=5, p38: n=3, PTP1B: n=7, Thrombin: n=3, Tyk2: n=6). |
| Data exclusions | SMILES that failed during preparation with RDKit were removed. Binding affinity measurements without values as well as uncertain, i.e., qualified data with either the "<" or ">" sign, were discarded. |
| Replication | To reproduce the primary results of this research, refer to the analytical pipeline available at https://doi.org/10.5281/zenodo.8275244. All experimental results can be successfully reproduced. |
| Randomization | 1) In the selection experiments, we initialized the model using random seeds 0 to 6. Randomness in the AI model training process affects the training of the model, which is unavoidable. In this experiment we set up the random seed in order to control the randomness and ensure that the relevant experimental results can be reproduced.<br> 2) In the model ranking performance evaluation, we randomly conducted 10 independent runs in every experiment. Since our model requires a reference molecule, the choice of the reference molecule affects the performance of the model. In order to more fully validate the model performance, we conducted 10 independent experiments. The reason for setting n=10 is that there is one test series which contains only 11 molecules and n=10 is sufficient to make a full assessment of the model performance. |
| Blinding | We were blinded to the group allocation during data collection and analysis.The group allocation process was performed by computer script without any manual intervention. |

# Reporting for specific materials, systems and methods

We require information from authors about some types of materials, experimental systems and methods used in many studies. Here, indicate whether each material, system or method listed is relevant to your study. If you are not sure if a list item applies to your research, read the appropriate section before selecting a response.

## Materials & experimental systems

| n/a | Involved in the study |
|-----|----------------------|
| ☒ | Antibodies |
| ☒ | Eukaryotic cell lines |
| ☒ | Palaeontology and archaeology |
| ☒ | Animals and other organisms |
| ☒ | Clinical data |
| ☒ | Dual use research of concern |

## Methods

| n/a | Involved in the study |
|-----|----------------------|
| ☒ | ChIP-seq |
| ☒ | Flow cytometry |
| ☒ | MRI-based neuroimaging |

