## [Peer Review File · Nature Computational Science]

Peer Review Information

Journal: Nature Computational Science

Manuscript Title: Computing Relative Binding Affinity of Ligands Based on a Pairwise Binding Comparison Network

Corresponding author name(s): Professor Mingyue Zheng

Reviewer Comments & Decisions:

Decision Letter, initial version:

Date: 7th July 23 13:07:57

Last Sent: 7th July 23 13:07:57

Triggered By: Kaitlin McCardle

From: kaitlin.mccardle@us.nature.com

To: myzheng@simm.ac.cn

Subject: Decision on Nature Computational Science manuscript NATCOMPUTSCI-23-0512

Message: ** Please ensure you delete the link to your author homepage in this e-mail if you wish to forward it to your co-authors. **

Dear Dr Zheng,

Your manuscript "PBCNet : Computing Relative Binding Affinity of Ligands Based on a Pairwise Binding Comparison Network" has now been seen by 2 referees, whose comments are appended below. You will see that while they find your work of interest, they have raised points that need to be addressed before we can make a decision on publication.

The referees' reports seem to be quite clear. Naturally, we will need you to address all of the points raised.

Please use the following link to submit your revised manuscript and a point-by-point response to the referees' comments (which should be in a separate document to any cover letter):

[REDACTED]

** This url links to your confidential homepage and associated information about manuscripts you may have submitted or be reviewing for us. If you wish to forward this e-mail to co-authors, please delete this link to your homepage first. **

To aid in the review process, we would appreciate it if you could also provide a copy of your manuscript files that indicates your revisions by making use of Track Changes or similar mark-up tools. Please also ensure that all correspondence is marked with your Nature Computational Science reference number in the subject line.

In addition, please make sure to upload a Word Document or LaTeX version of your text, to assist us in the editorial stage.

If you have any issues when updating your Code Ocean capsule during the revision process, please email the Code Ocean support team Cc'ing me.

To improve transparency in authorship, we request that all authors identified as 'corresponding author' on published papers create and link their Open Researcher and Contributor Identifier (ORCID) with their account on the Manuscript Tracking System (MTS), prior to acceptance. ORCID helps the scientific community achieve unambiguous attribution of all scholarly contributions. You can create and link your ORCID from the home page of the MTS by clicking on 'Modify my Springer Nature account'. For more information please visit www.springernature.com/orcid.

We hope to receive your revised paper within three weeks. If you cannot send it within this time, please let us know.

Best regards,

Kaitlin McCardle, PhD
Associate Editor
Nature Computational Science

Reviewers comments:

Reviewer #1 (Remarks to the Author):

This manuscript introduces a new deep learning model PBCnet for predicting the relative binding free energies (RBF) of protein-ligand complexes for lead optimization. The test results show that PBCnet performs significantly better than docking scoring and other deep learning models in and is slightly better than the physics based MM-GBSA. Their method is still less accurate than the most rigorous and much more computationally costly FEP+ method. While this work is potentially of considerable interest to the readers of the Nature Computational Science, I have some questions regarding the manuscript:

1. How robust is the accuracy of the prediction results in the presence of the small variations in the structures of the input protein-ligand complexes? Say if the bound ligand reorients by about 0.5 Å in terms of RMSD from the original input structure, how does this small change affect the predicted ranking of the RBF? Such tests are relevant in real applications since even the best docking poses cannot be perfectly

reproduce the crystal structures and typical acceptable deviations is $\text{RMSD} < 2.0 \text{ \AA}$.
2. Table 1 and Table 2. How are the results reported using other methods obtained? If they are all calculated in this work, then the details of the calculations using these other methods need to be included in the Methods. For example, all the parameters used in the MM-GBSA calculations, etc.
3. In their simulation experiment, the PBCNet speeds up the lead optimization campaigns by 30%, which is less impressive. How does this number compare with similar tests using say Glide docking or MM-GBSA or one of the other deep learning models?

Reviewer #2 (Remarks to the Author):

The paper by Yu et al. introduces the pairwise binding comparison network (PBCNet) as a solution to the ongoing challenge of structure-based lead optimization in drug discovery. PBCNet utilizes a specialized physics-informed graph attention mechanism to effectively evaluate and rank the relative binding affinity of closely related ligands. Extensive benchmarking against independent ligand and target sets convincingly establishes PBCNet's predictive accuracy and computational efficiency compared to existing methods. Notably, PBCNet achieves comparable performance to the computationally intensive FEP method while eliminating the need for extensive expert involvement through the incorporation of a fine-tuning operation. The authors further demonstrate that by leveraging active learning optimization in conjunction with PBCNet, lead optimization campaigns can be expedited by an impressive 30%. To enhance accessibility, the authors have developed a user-friendly web service, which enables researchers to conveniently predict complex relative binding affinities using PBCNet's intuitive graphical interface. I found their work exceptionally interesting.

However, before the paper can be considered for publication in Nature Computer Science, I suggest the authors address the following points:

1. Provide further insights into the underlying mechanisms of PBCNet by analyzing how the graph attention mechanism captures key features and interactions between ligands and target proteins.
2. Enhance the interpretability of PBCNet's predictions by providing insights into the features or structural motifs that contribute most significantly to the binding affinity rankings. This could help medicinal chemists gain a better understanding of the underlying factors driving the predictions and guide their decision-making process.
3. While the authors convincingly demonstrate comparable performance to FEP calculations, they did not investigate the decline in PBCNet's predictive power with increasing structural differences between reference and test ligands. This aspect should be explored and discussed.
4. The method's reliance on the availability of a protein/reference ligand complex should be elaborated upon, distinguishing between experimental and theoretical structures and their potential impact on the results.
5. Regarding the provided web server, I observed that it works well when uploading a protein structure, reference ligand, and test ligand files. However, it would be advantageous if multiple test ligand files could be uploaded simultaneously to streamline the server's usability.

Author Rebuttal to Initial comments**Reviewer #1:**

This manuscript introduces a new deep learning model PBCNet for predicting the relative binding free energies (RBFE) of protein-ligand complexes for lead optimization. The test results show that PBCNet performs significantly better than docking scoring and other deep learning models in and is slightly better than the physics-based MM-GBSA. Their method is still less accurate than the most rigorous and much more computationally costly FEP+ method. While this work is potentially of considerable interest to the readers of the Nature Computational Science, I have some questions regarding the manuscript:

Comment 1: How robust is the accuracy of the prediction results in the presence of the small variations in the structures of the input protein-ligand complexes? Say if the bound ligand reorients by about 0.5 Å in terms of RMSD from the original input structure, how does this small change affect the predicted ranking of the RBFE? Such tests are relevant in real applications since even the best docking poses cannot be perfectly reproduce the crystal structures and typical acceptable deviations is $\text{RMSD} < 2.0 \text{ \AA}$.

Reply:

We greatly appreciate the valuable comments provided by the reviewer. As suggested, we investigated the impact of small changes in ligand poses on the predictive ability of PBCNet, using the FEP1 set for the analysis.

First, for each PDB in the FEP1 set, we redocked co-crystallized ligands to their corresponding proteins using Glide SP with default parameters. The docking poses with RMSDs less than 1.0 Å were retained as biased poses, which are all slightly different from the experimentally determined poses. Then, the biased poses of the remaining compounds were generated for each system through the shape-constrained docking process (Schrödinger 2020-4), where the biased poses of the co-crystallized ligand were sequentially used as shape references. Through these operations, we constructed the biased FEP1 set (Table S4), where each system has several biased poses.

To evaluate the ranking performance of PBCNet on the biased FEP1 set, we used the same protocol as described in the manuscript (Section 2.2.1), and the mean values and variances of the ranking metrics are summarized in Table 5.

Table 5 shows that the ranking performance of PBCNet on the biased FEP1 set is very similar to that on the original FEP1 set (Pearson: 0.64 vs 0.65, Spearman: 0.64 vs 0.64). This result clearly demonstrated that our model is highly robust to small changes in ligand poses. One potential reason for this robustness is that the ligand poses used for model training were produced by molecular docking, as explained in Section 4.2.1 of the manuscript. Some variance has been introduced in the binding pose generation procedures, which may be considered as a data augmentation. We thank the reviewer for raising this important point, and we have included the analysis in the revised manuscript.

Table S4. The statistical table of the biased poses of each system in FEP1 set

Systems	Number of biased series	RMSDs
Bace	7	0.189 Å, 0.218 Å, 0.242 Å, 0.27 Å, 0.278 Å, 0.290 Å, 0.372 Å
CDK2	3	0.229 Å, 0.354 Å, 0.5 Å
JNK1	3	0.422 Å, 0.499 Å, 0.535 Å
MCL1	5	0.108 Å, 0.356 Å, 0.425 Å, 0.568 Å, 0.597 Å
p38	3	0.544 Å, 0.835 Å, 0.768 Å
PTP1B	7	0.627 Å, 0.641 Å, 0.7 Å, 0.708 Å, 0.710 Å, 0.774 Å, 0.861 Å
Thrombin	3	0.191 Å, 0.255 Å, 0.453 Å
Tyk2	6	0.146 Å, 0.22 Å, 0.622 Å, 0.658 Å, 0.736 Å, 0.96 Å

Table 5. The ranking performance of PBCNet on the original and biased FEP1 sets ^a

		BACE	CDK2	JNK1	MCL1	p38	PTP1B	Thrombin	Tyk2	Average
Original poses	Pearson	0.61	0.66	0.41	0.72	0.56	0.75	0.84	0.64	0.65
	Spearman	0.52	0.66	0.47	0.73	0.56	0.71	0.82	0.61	0.64
Biased poses (Poses with deviations)	Pearson	0.61 (1.1×10^{-4})	0.69 (8.8×10^{-5})	0.43 (2.9×10^{-5})	0.66 (3.0×10^{-4})	0.61 (2.6×10^{-4})	0.73 (8.8×10^{-4})	0.83 (5.3×10^{-5})	0.55 (2.0×10^{-3})	0.64
	Spearman	0.58 (1.0×10^{-3})	0.69 (1.6×10^{-4})	0.48 (2.1×10^{-4})	0.69 (3.2×10^{-4})	0.62 (6.5×10^{-4})	0.74 (6.2×10^{-4})	0.84 (8.2×10^{-5})	0.51 (1.9×10^{-3})	0.64

a. For each system in the biased FEP1 set, the average and the variance (in brackets) of the ranking metrics based on different biased poses are all reported.

Comment 2: Table 1 and Table 2. How are the results reported using other methods obtained? If they are all calculated in this work, then the details of the calculations using these other methods need to be included in the Methods. For example, all the parameters used in the MM-GBSA calculations, etc.

Reply:

The performance metrics of all baseline models in Table 1 and Table 2, except PIGNet, were taken from the following papers:

1. FEP+, Glide SP, and MM-GB/SA in Table 1: Wang, L. et al. Accurate and reliable prediction of relative ligand binding potency in prospective drug discovery by way of a modern free-energy calculation protocol and force field. *Journal of the American Chemical Society* **2015**, 137, 2695-2703.
2. FEP+, Glide SP, and MM-GB/SA in Table 2: Schindler, C. E. M. et al. Large-scale assessment of binding free energy calculations in active drug discovery projects. *Journal of Chemical Information and Modeling* **2020**, 60, 5457-5474.
3. DeltaDelta: Jiménez-Luna, J. et al. DeltaDelta neural networks for lead optimization of small molecule potency. *Chemical Science* **2019**, 10, 10911-10918.
4. Default2018 and Dense: McNutt, A. T. & Koes, D. R. Improving $\Delta\Delta G$ Predictions with a Multitask Convolutional Siamese Network. *Journal of Chemical Information and Modeling* **2022**, 62, 1819-1829.

For PIGNet, the results were calculated using its officially reported code and weights. The revised manuscript now includes this information as per the reviewer's suggestion.

Comment 3: In their simulation experiment, the PBCNet speeds up the lead optimization campaigns by 30%, which is less impressive. How does this number compare with similar tests using say Glide docking or MM-GBSA or one of the other deep learning models?

Reply:

We appreciate the reviewer's comment, but the performance metric "Advantage ratio" does not indicate the proportion of lead optimization projects accelerated by PBCNet. Accordingly, we have introduced a new metric called the "Efficiency improvement ratio", defined as follows:

$$\text{Efficiency improvement ratio} = \frac{\text{Experimental order} - \text{Model selection order}}{\text{Model selection order}} \times 100\%$$

This metric represents the efficiency increase of completing a compound optimization project before and after using PBCNet, assuming that a project ends once the most active compound is obtained. As shown in revised Table 3, PBCNet achieved an impressive average efficiency increase of 473%.

As recommended by the reviewer, we conducted an analysis using MM-GB/SA as a baseline for comparison. MM-GB/SA was implemented using the Schrödinger Prime MM-GBSA with default settings, and the results are presented in Table S3. Our findings demonstrated that

PBCNet consistently outperforms MM/GB-SA across all evaluated metrics (Advantage ratio: 30.18% vs 14.93%, Efficiency improvement ratio: 473.0% vs 281.0%).

We appreciate the reviewer for pointing out these confusing points, and we have incorporated the added analysis into the revised manuscript.

Table 3. Selection experiment results of the active learning equipped PBCNet for 9 different datasets

System	Number of ligands	Experimental order ^a	Model selection order ^b ($\beta = 2$)	Model selection order ^b ($\beta = -2$)	Model selection order ^b ($\beta = 0$)	Advantage order ^c ($\beta = 2$)	Advantage ratio ^d ($\beta = 2$)	Efficiency improvement ratio ($\beta = 2$)
FGFR2	15	7	5±0	6±0	6±0	2	13.33%	40.0%
BCL6	25	23	2	2	2	21	84.00%	1050.0%
HO1	19	8	9.33±0.47	9±0	9±0	-1.33	-7.00%	-14.0%
LRRK2	20	6	15±0	15±0	15±0	-9	-45.00%	-60.0%
sEH	51	43	2	2	2	41	80.39%	2050.0%
CKD9	38	13	15.17±1.34	15.83±2.97	18.33±2.69	-2.17	-7.75%	-14.0%

WDR5	21	16	12.83±1.87	12.83±1.21	13.83±0.90	3.17	15.10%	25.0%
AAK1	30	28	3	3	3	25	83.33%	833.33%
PSK13	38	27	6±0	5±0	5±0	21	55.26%	350.0%
mean	28.56	19	7.81±0.23	7.85±0.31	8.35±0.29	11.18	30.18%	473.0%

a. The order of the target compound (the compound with the highest affinity in each chemical series) reported in the original literature.

b. The average and corresponding variance values based on six independently runs with different random seeds are reported here.

Table S3. Selection experiment results of the MM/GB-SA for 9 different datasets

System	Number of ligands	Experimental order ^a	MM/GB-SA order	Advantage order ^c	Advantage ratio ^d	Efficiency improvement ratio
FGFR2	15	7	4	3	20.0%	75.0%
BCL6	25	23	1	22	88.0%	2200.0%

HO1	19	8	9	-1	-5.26%	11.0%
LRRK2	20	6	14	-8	-40.0%	57.0%
sEH	51	43	11	32	62.75%	291.0%
CKD9	38	13	17	-4	-10.53%	24.0%
WDR5	21	16	18	-2	-9.52%	11.0%
AAK1	30	28	28	0	0.0%	0.0%
PSK13	38	27	16	11	28.95%	69.0%
mean	28.56	19	13.11	5.89	14.93%	281.0%

- The order of the target compound (the compound with the highest affinity in each chemical series) reported in the original literature.
- The average and corresponding variance values based on six independently runs with different random seeds are reported here.

Reviewer #2:

The paper by Yu et al. introduces the pairwise binding comparison network (PBCNet) as a solution to the ongoing challenge of structure-based lead optimization in drug discovery. PBCNet utilizes a specialized physics-informed graph attention mechanism to effectively evaluate and rank the relative binding affinity of closely related ligands. Extensive benchmarking against independent ligand and target sets convincingly establishes PBCNet's predictive accuracy and computational efficiency compared to existing methods. Notably, PBCNet achieves comparable performance to the computationally intensive FEP method while eliminating the need for extensive expert involvement through the incorporation of a fine-tuning operation. The authors further demonstrate that by leveraging active learning optimization in conjunction with PBCNet, lead optimization campaigns can be expedited by an impressive 30%.

To enhance accessibility, the authors have developed a user-friendly web service, which enables researchers to conveniently predict complex relative binding affinities using PBCNet's intuitive graphical interface. I found their work exceptionally interesting.

However, before the paper can be considered for publication in Nature Computer Science, I suggest the authors address the following points:

Comment 1: Provide further insights into the underlying mechanisms of PBCNet by analyzing how the graph attention mechanism captures key features and interactions between ligands and target proteins.

Reply:

We would like to highlight two important features of PBCNet's graph attention mechanism: 1) the alternating updates of edge and node information; 2) the integration of 3D geometric information.

In most graph convolution models, nodes are considered primary units of information propagation, while edges are used for connectivity and provide minimal chemical information. However, in PBCNet's message passing phase, both edges and nodes are given equal importance. This approach enables our model to pay more attention to edge information, which

allows it to identify potential inter-molecular interactions between protein and ligand atom pairs. The message passing of PBCNet consists of two phases: (1) distance and angle-aware bond-to-bond blocks (DAEE blocks); (2) distance-aware bond-to-atom blocks (DEN blocks). Both modules use neighboring elements' information to update their representations, which effectively captures the contextual environment in which they are embedded. Section 4.1 of our manuscript provides a comprehensive explanation of the message passing process, including mathematical formulae and an in-depth discussion of our model's attention mechanism.

Since inter-molecular interactions follow strict geometric rules, such as hydrogen bonds, we introduce two physically oriented modelling strategies: spatial distance and angle geometric information. These strategies help the model better capture key inter-molecular interactions in the modelling process. The introduction of distance and angle information transforms the original 2D molecular graph into a 3D molecular graph, which more comprehensively represents the binding information between the protein and ligand. We encode distance and angle information and incorporate them as attention biases into PBCNet's attention mechanism.

Our ablation studies confirm that the incorporation of geometric information significantly enhances PBCNet's predictive performance. Further atom-level interpretability analysis shows that our model assigns higher attention scores to atom pairs involved in inter-molecular interactions. We also conducted a substructure-level interpretability analysis and found that substructures forming inter-molecular interactions have a greater influence on the model's predictions. These experimental results demonstrate the soundness of PBCNet's design and the validity of our modeling strategies.

Comment 2: Enhance the interpretability of PBCNet's predictions by providing insights into the features or structural motifs that contribute most significantly to the binding affinity rankings. This could help medicinal chemists gain a better understanding of the underlying factors driving the predictions and guide their decision-making process.

Reply:

We greatly appreciate the valuable comments provided by the reviewer. As suggested, we conducted a novel interpretability analysis experiment at the substructure level on PBCNet, to provide medicinal chemists with a better understanding of PBCNet's predictions.

In this analysis, we employed the Substructure Mask Explanation (SME) methodology, as recently proposed by Wu et al. (*Nat Commun* **2023**, 14, 2585). We assume that the model's prediction value for a compound is denoted as \hat{y} . Then, the compounds are split into substructures using the BRICS method. Sequentially, the hidden representations of the atoms of each substructure are masked during the model's readout phase, yielding the corresponding model prediction value \hat{y}_{sub_i} where the subscript sub_i represents the i^{th} substructure. When the predicted value represents the compound's activity, we consider that a greater decrease in \hat{y}_{sub_i} compared to \hat{y} indicates that the corresponding substructure plays a more crucial role in the model's prediction. Thus, the attribution scores used to quantify the importance of each substructure are defined by the following equation:

$$\text{Attribution}_{sub_i} = \hat{y} - \hat{y}_{sub_i}$$

and we normalize the attribution scores to normalized attribution scores (Attribution_N) within a range of 0 and 1, as per the following equation:

$$\text{Attribution_N}_{sub_i} = \frac{\text{Attribution}_{sub_i}}{\sum_{i=1}^N \text{Attribution}_{sub_i}}$$

where N is the number of the substructures.

Here, we take compound 6a from the Thrombin system as a case study, using compound 1a as a reference ligand to illustrate PBCNet's activity prediction for compound 6a (Figure 5C). Compound 6a was segmented into 7 substructures by BRICS method, with the amide group being divided into two distinct substructures. To provide a more intuitive representation for medicinal chemists, we manually merged the amide group as a whole (Figure 5A). The corresponding results were summarized in Figure 5A, and the visualization is presented in Figure 5B.

As shown, we found that Sub₄ and Sub₁ have the greatest impact on the predictive results. PBCNet is designed to predict the relative binding affinities, which predominantly derived from

the different substructures of a pair of ligands. Sub₄, being the part of compound 6a that structurally deviates from compound 1a, has been emphasized, suggesting that PBCNet indeed captures the structural differences between input ligands. Moreover, as depicted in Figure 4A, Sub₁ forms two hydrogen bonds with the protein, so the emphasizing of Sub₁ also implies that PBCNet focuses on key molecular motifs that form inter-molecular interactions.

We have incorporated the analysis into the revised manuscript.

Figure 5. The result of PBCNet’s interpretability analysis on the substructure level. **A.** Structure and predict activity relationship of a series of Thrombin inhibitors, as well as the attribution score of each substructure (highlighted); **B.** The visualization of the analysis, and the numbers on the subgraph indicate the normalized attribution scores of the corresponding substructures; **C.** The binding modes of compound 6a (cyan) and 1a (purple) within the protein pocket.

A. Thrombin: 6a

Type	Ligand Atom	Protein Atom	Protein Atom remove Hs
H-bond	O 3	H: GLY216	N: GLY216
H-bond	N 8	O: SER214	-
H-bond	N 10	O: GLY216	-

B. JNK1: 18660-1

Type	Ligand Atom	Protein Atom	Protein Atom remove Hs
H-bond	N 12	O: GLU109	-
H-bond	O 18	H: MET111	N: MET111
Negative	C 5	-	-

Figure 4. Interpretability analysis results of PBCNet on two ligands: **A.** a thrombin inhibitor 6a and **B.** a JNK1 inhibitor 18660-1. The molecular structure, hydrogen-bond calculation results, 3-dimensional hydrogen-bonds visualization graphs, and attention visualization graphs are shown for comparison. In each attention visualization graph, the ligand atom (referred to as target atom) is denoted by a red dot, indicated by an arrow, and is involved in the formation of hydrogen bonds. Other dots denote the neighbor atoms of the target atom. The black ones represent the ligand atoms (including the virtual aromatic nodes in the ligand structure) covalently linked with the target atom, the grey ones represent the protein pocket atoms (including the virtual aromatic nodes in the protein structure) linked with the target atom by virtual distance edges, and the dot in cyan denotes the protein pocket atom that form the hydrogen-bond with the target atom. The color depth of the edges indicates the attention score, and an edge with a dark color is favorable for protein-ligand binding.

Comment 3: While the authors convincingly demonstrate comparable performance to FEP calculations, they did not investigate the decline in PBCNet's predictive power with increasing structural differences between reference and test ligands. This aspect should be explored and discussed.

Reply:

We would like to thank the reviewer for this suggestion. The issue primarily pertains to the applicability domain of the model. PBCNet is specifically designed to infer the activity differences of structural analogues during lead optimization. In such cases, the molecules generally share structural similarities, and our training set comprises the molecule pairs with Tanimoto similarity scores higher than 0.6. To assess the correlation between the model accuracy and molecule similarity, we segmented all pairwise samples from both test sets into five bins, ordered by Tanimoto similarity scores, and computed the mean absolute errors (MAEs) for each bin. As shown in Figure 2A, we observed a negative correlation between the ligand similarity and MAE.

We further analyzed the proportions of ligand pairs with different similarity scores in the FEP1 and FEP2 sets (Figure 2B). Figure 2B demonstrates that a significant proportion (70.4%) of molecule pairs in the FEP2 set have Tanimoto similarity scores below 0.6. However, PBCNet's ranking performance on the FEP2 set still surpassed all the baselines, except for FEP+

(Table 2 in our manuscript). Given the consistent performance on FEP1 and FEP2, we may conclude that PBCNet should be of practical value for guiding lead optimization projects.

Figure 2 A. The bar plot shows the change of model accuracy with the pairwise molecule similarity. We split all pairwise samples in both test sets, ordered by Tanimoto similarity scores in five bins (x-axis), and calculated the mean absolute errors (MAEs) for each bin (y-axis). The error bars represent 0.1 times standard deviation (SD). **B.** The bar plot shows the proportions of ligand pairs (y-axis) with different Tanimoto similarity scores (x-axis) in the FEP1 and FEP2 sets. The proportion of molecules pairs with a Tanimoto similarity score less than 0.6 in the FEP2 set are significantly higher than that in the FEP1 set (70.4% vs 54.4%), and all pairs with a Tanimoto similarity score less than 0.2 are from the FEP2 set.

Comment 4: The method's reliance on the availability of a protein/reference ligand complex should be elaborated upon, distinguishing between experimental and theoretical structures and their potential impact on the results.

Reply:

We greatly appreciate the suggestion from the reviewer. Reviewer #1 also made a similar comment about the impact of small changes in ligand poses on the predictive ability of PBCNet. The reply to Reviewer #1 is provided below for reference.

We used the FEP1 set for the analysis. First, for each PDB in the FEP1 set, we redocked co-crystallized ligands to their corresponding proteins using Glide SP with default parameters. The docking poses with RMSDs less than 1.0 Å were retained as biased poses, which are all slightly different from the experimentally determined poses. Then, the biased poses of the remaining

compounds were generated for each system through the shape-constrained docking process (Schrödinger 2020-4), where the biased poses of the co-crystallized ligand were sequentially used as shape references. Through these operations, we constructed the biased FEP1 set (Table S4), where each system has several biased poses.

To evaluate the ranking performance of PBCNet on the biased FEP1 set, we used the same protocol as described in the manuscript (Section 2.2.1), and the mean values and variances of the ranking metrics are summarized in Table 5.

Table 5 shows that the ranking performance of PBCNet on the biased FEP1 set is very similar to that on the original FEP1 set (Pearson: 0.64 vs 0.65, Spearman: 0.64 vs 0.64). This result clearly demonstrated that our model is highly robust to small changes in ligand poses. One potential reason for this robustness is that the ligand poses used for model training were produced by molecular docking, as explained in Section 4.2.1 of the manuscript. Some variance has been introduced in the binding pose generation procedures, which may be considered as a data augmentation. We thank the reviewer for raising this important point, and we have included the analysis in the revised manuscript.

Table S4. The statistical table of the biased poses of each system in FEP1 set

Systems	Number of biased series	RMSDs
Bace	7	0.189 Å, 0.218 Å, 0.242 Å, 0.27 Å, 0.278 Å, 0.290 Å, 0.372 Å
CDK2	3	0.229 Å, 0.354 Å, 0.5 Å
JNK1	3	0.422 Å, 0.499 Å, 0.535 Å
MCL1	5	0.108 Å, 0.356 Å, 0.425 Å, 0.568 Å, 0.597 Å
p38	3	0.544 Å, 0.835 Å, 0.768 Å
PTP1B	7	0.627 Å, 0.641 Å, 0.7 Å, 0.708 Å, 0.710 Å, 0.774 Å, 0.861 Å
Thrombin	3	0.191 Å, 0.255 Å, 0.453 Å
Tyk2	6	0.146 Å, 0.22 Å, 0.622 Å, 0.658 Å, 0.736 Å, 0.96 Å

Table 5. The ranking performance of PBCNet on the original and biased FEP1 sets ^a

		BACE	CDK2	JNK1	MCL1	p38	PTP1B	Thrombin	Tyk2	Average
Original poses	Pearson	0.61	0.66	0.41	0.72	0.56	0.75	0.84	0.64	0.65
	Spearman	0.52	0.66	0.47	0.73	0.56	0.71	0.82	0.61	0.64

Biased poses (Poses with deviations)	Pearson	0.61 (1.1×10^{-4})	0.69 (8.8×10^{-5})	0.43 (2.9×10^{-5})	0.66 (3.0×10^{-4})	0.61 (2.6×10^{-4})	0.73 (8.8×10^{-4})	0.83 (5.3×10^{-5})	0.55 (2.0×10^{-3})	0.64
	Spearman	0.58 (1.0×10^{-3})	0.69 (1.6×10^{-4})	0.48 (2.1×10^{-4})	0.69 (3.2×10^{-4})	0.62 (6.5×10^{-4})	0.74 (6.2×10^{-4})	0.84 (8.2×10^{-5})	0.51 (1.9×10^{-3})	0.64

- a. For each system in the biased FEPI set, the average and the variance (in brackets) of the ranking metrics based on different biased poses are all reported.

Comment 5: Regarding the provided web server, I observed that it works well when uploading a protein structure, reference ligand, and test ligand files. However, it would be advantageous if multiple test ligand files could be uploaded simultaneously to streamline the server's usability.

Reply:

As per the reviewer's suggestion, we have added a new feature that allows users to upload multiple test ligand files (up to 10) simultaneously. This update should improve the user experience and streamline the process.

Decision Letter, first revision:

Date: 11th August 23 12:30:30

Last Sent: 11th August 23 12:30:30

Triggered By: Kaitlin McCardle

From: kaitlin.mccardle@us.nature.com

To: myzheng@simm.ac.cn

CC: computacionalscience@nature.com

Subject: AIP Decision on Manuscript NATCOMPUTSCI-23-0512A

Message: Our ref: NATCOMPUTSCI-23-0512A

11th August 2023

Dear Dr. Zheng,

Thank you for submitting your revised manuscript "PBCNet : Computing Relative Binding Affinity of Ligands Based on a Pairwise Binding Comparison Network" (NATCOMPUTSCI-23-0512A). It has now been seen by the original referees and their

comments are below. The reviewers find that the paper has improved in revision, and therefore we'll be happy in principle to publish it in Nature Computational Science, pending minor revisions to comply with our editorial and formatting guidelines.

TRANSPARENT PEER REVIEW

Nature Computational Science offers a transparent peer review option for original research manuscripts. We encourage increased transparency in peer review by publishing the reviewer comments, author rebuttal letters and editorial decision letters if the authors agree. Such peer review material is made available as a supplementary peer review file. **Please remember to choose, using the manuscript system, whether or not you want to participate in transparent peer review.**

Thank you again for your interest in Nature Computational Science. Please do not hesitate to contact me if you have any questions.

Sincerely,

Kaitlin McCardle, PhD
Associate Editor
Nature Computational Science

ORCID

Reviewer #1 (Remarks to the Author):

All my questions/comments have been adequately addressed in the revised manuscript. I now recommend its publication as is.

Reviewer #2 (Remarks to the Author):

The authors have sufficiently addressed all the issues raised by this reviewer. The manuscript can be now published in the Nature Computational Science journal.

Final Decision Letter:

Date: 5th September 23 10:53:41
Last Sent: 5th September 23 10:53:41
Triggered By: Fernando Chirigati
From: fernando.chirigati@us.nature.com
To: myzheng@simm.ac.cn
CC: kaitlin.mccardle@us.nature.com
BCC: fernando.chirigati@us.nature.com,rjsproduction@springernature.com,computationalscience@nature.com,rjsart@springernature.com
Subject: Decision on Nature Computational Science manuscript NATCOMPUTSCI-23-0512B
Message: Dear Professor Zheng,

:
 We are pleased to inform you that your Article "Computing Relative Binding Affinity of Ligands Based on a Pairwise Binding Comparison Network" has now been accepted for publication in Nature Computational Science.

Once your manuscript is typeset, you will receive an email with a link to choose the appropriate publishing options for your paper and our Author Services team will be in touch regarding any additional information that may be required.

Please note that *Nature Computational Science* is a Transformative Journal (TJ). Authors may publish their research with us through the traditional subscription access route or make their paper immediately open access through payment of an article-processing charge (APC). Authors will not be required to make a final decision about access to their article until it has been accepted. [Find out more about Transformative Journals](https://www.springernature.com/gp/open-research/transformative-journals)

Authors may need to take specific actions to achieve [compliance with funder and institutional open access mandates](https://www.springernature.com/gp/open-research/funding/policy-compliance-faqs). If your research is supported by a funder that requires immediate open access (e.g. according to [Plan S principles](https://www.springernature.com/gp/open-research/plan-s-compliance)) then you should select the gold OA route, and we will direct you to the compliant route where possible. For authors selecting the subscription publication route, the journal's standard licensing terms will need to be accepted, including [journal](https://www.springernature.com/gp/open-research/policies/journal-)

policies">self-archiving policies. Those licensing terms will supersede any other terms that the author or any third party may assert apply to any version of the manuscript.

Acceptance of your manuscript is conditional on all authors' agreement with our publication policies (see <https://www.nature.com/natcomputsci/for-authors>). In particular your manuscript must not be published elsewhere and there must be no announcement of the work to any media outlet until the publication date (the day on which it is uploaded onto our web site).

Before your manuscript is typeset, we will edit the text to ensure it is intelligible to our wide readership and conforms to house style. We look particularly carefully at the titles of all papers to ensure that they are relatively brief and understandable.

Once your manuscript is typeset, you will receive a link to your electronic proof via email with a request to make any corrections within 48 hours. If, when you receive your proof, you cannot meet this deadline, please inform us at rjsproduction@springernature.com immediately.

If you have queries at any point during the production process then please contact the production team at rjsproduction@springernature.com. Once your paper has been scheduled for online publication, the Nature press office will be in touch to confirm the details.

Content is published online weekly on Mondays and Thursdays, and the embargo is set at 16:00 London time (GMT)/11:00 am US Eastern time (EST) on the day of publication. If you need to know the exact publication date or when the news embargo will be lifted, please contact our press office after you have submitted your proof corrections. Now is the time to inform your Public Relations or Press Office about your paper, as they might be interested in promoting its publication. This will allow them time to prepare an accurate and satisfactory press release. Include your manuscript tracking number NATCOMPUTSCI-23-0512B and the name of the journal, which they will need when they contact our office.

About one week before your paper is published online, we shall be distributing a press release to news organizations worldwide, which may include details of your work. We are happy for your institution or funding agency to prepare its own press release, but it must mention the embargo date and Nature Computational Science. Our Press Office will contact you closer to the time of publication, but if you or your Press Office have any inquiries in the meantime, please contact press@nature.com.

We welcome the submission of potential cover material (including a short caption of around 40 words) related to your manuscript; suggestions should be sent to Nature

Computational Science as electronic files (the image should be 300 dpi at 210 x 297 mm in either TIFF or JPEG format). We also welcome suggestions for the Hero Image, which appears at the top of our [home page](http://www.nature.com/natcomputsci); these should be 72 dpi at 1400 x 400 pixels in JPEG format. Please note that such pictures should be selected more for their aesthetic appeal than for their scientific content, and that colour images work better than black and white or grayscale images. Please do not try to design a cover with the Nature Computational Science logo etc., and please do not submit composites of images related to your work. I am sure you will understand that we cannot make any promise as to whether any of your suggestions might be selected for the cover of the journal.

Best,
Fernando (on behalf of Kaitlin McCardle)

--

Fernando Chirigati, PhD
Chief Editor, Nature Computational Science
Nature Portfolio

P.S. Click on the following link if you would like to recommend Nature Computational Science to your librarian: <https://www.springernature.com/gp/librarians/recommend-to-your-library>

** Visit the Springer Nature Editorial and Publishing website at <http://editorial-jobs.springernature.com> for more information about our career opportunities. If you have any questions please click [here](mailto:editorial.publishing.jobs@springernature.com).**